



**Carbon turnover in cell compartments and microbial groups in soil**
Anna Gunina[1,2], Michaela Dippold[1], Bruno Glaser[3], Yakov Kuzyakov[1,4]
[1] Department of Agricultural Soil Science, Georg-August-University of Göttingen,
Büsgenweg 2, 37077 Göttingen, Germany
[2] Department of Soil Biology and Biochemistry, Dokuchaev Soil Science Institute, Russian
Federation;
[3] Department of Soil Biogeochemistry, Institute of Agricultural and Nutritional Science,
Martin-Luther University Halle-Wittenberg, von-Seckendorff-Platz 3, 06120 Halle (Saale),
Germany
[4] Department of Soil Science of Temperate Ecosystems, Georg-August-University of
Göttingen, Büsgenweg 2, 37077, Göttingen, Germany
Corresponding Author:
Anna Gunina
Department of Agricultural Soil Science
Georg-August-University of Göttingen
Büsgenweg 2
37077 Göttingen
Tel: 0551/39-20502
email:  guninaann@gmail.com
Tel.: 0157/85566093





**Abstract**
Microorganisms regulate the carbon (C) cycle in soil, controlling the utilization and
recycling of organic substances. To reveal the contribution of particular microbial groups to
C utilization and C turnover within the microbial cells, fate of $^{13}$C-labeled glucose was
studied under field conditions. The $^{13}$C was traced in cytosolic substances, amino sugars and
phospholipid fatty acids (PLFA) at intervals of 3, 10 and 50 days after glucose addition.

$^{13}$C enrichment into PLFA (~1.5% of PLFA C at day 3) was one order of magnitude

greater than into the cytosol, showing the importance of cell membranes for initial C
utilization. $^{13}$C enrichment of amino sugars in living microorganisms at day 3 accounted for
0.57%, resulting that the turnover of cell wall components is two times slower than that of
cell membranes. Turnover time of C in the cytosol (150 days) was three times longer than in
PLFAs (47 days). Consequently, despite the lability of cytosol pool and expected fast
turnover rates, intensive recycling of cytosol components, within the living cells, leads to a
longer turnover time. Amino sugars originate mainly from microbial residues, thus longer
experimental periods are required for estimation of their turnover times.

Both PLFA and amino sugar profiles indicated that glucose C was preferentially used

by bacteria. The $^{13}$C incorporated into bacterial cell membrane components decreased with
time, but it remained constant or even increased for filamentous microorganisms. Hence, over
a short period, bacteria contribute more to the utilization of low molecular weight organic
substances, whereas filamentous microorganisms are responsible for further C
transformations. Thus, tracing $^{13}$C in cellular compounds with contrasting turnover rates
elucidated the role of microbial groups and their cellular compartments in C utilization and
recycling in soil. This information is especially important for assessing C fluxes in soil and
the contribution of C from microbial residues to soil organic matter.



**Keywords**
Microbial biomarkers; phospholipid fatty acids; amino sugars; $^{13}$C labeling; glucose
utilisation; soil microbial biomass.



## 1. Introduction

Over the last decade, numerous studies have demonstrated the role of soil microorganisms in regulating the fate and transformation of organic compounds. Soil microorganisms produce exoenzymes to carry out the primary degradation of plant as well as microbial polymers to monomers. Further transformations of monomers then take place within the microbial cells. Monomeric substances pass into the living microbial pool and are partly mineralised to $CO_2$, while part is assimilated into cell polymers and ultimately incorporated into soil organic matter (SOM) after cell death (Kindler et al., 2006). Understanding the fate of substances originated from plants and microbial residues into living biomass is therefore crucial for estimating the recycling of carbon (C) in soil and its stabilization as SOM.

Living microbial biomass (MB) is a highly active and heterogeneous pool, although it accounts for only 2-4% of the total SOM (Jenkinson and Ladd, 1981). Heterogeneity is evident at the level of single cells in the various cellular compartments with different properties, structures and biochemistry: from the highly heterogeneous cytosol (Malik et al., 2013), to well-structured cell membranes and cell walls. Due to their chemical composition and spatial localization, compounds of cell membranes (phospholipid fatty acids (PLFAs)) and cell walls (amino sugars) have different turnover times within the cell as well as different stabilities within SOM.

Organic compounds that are taken up by microorganisms first enter the cytosol (Bremer and Kuikman, 1994), which is presumed to be the most dynamic pool within microbial cells. However, due to the heterogeneity of this pool, no single C turnover time can be estimated. The calculated turnover time of intact PLFAs in soil after microbial death is 2.8 days (Kindler et al., 2009), resulting PLFAs are mainly used to characterize the living microorganisms (Frostegard et al., 2011; Rethemeyer, 2004). However, no data concerning turnover time of PLFA C in the living biomass are currently published. The formation of





amino sugars from plant biomass is relatively rapid at 6.2–9.0 days (Bai et al., 2013), whereas
their turnover times in soil vary between 6.5–81.0 $y^{-1}$ (Glaser et al., 2006). Thus, PLFAs and
amino sugars can be used to trace the fate of C within the living microorganisms and estimate
their contribution to SOM (Schmidt et al., 2007).

Some cell compartments, such as the cytoplasm, are not specific for various microbial

groups, whereas phospholipids are partly specific and consequently can be used to estimate
microbial community structure. Thus, PLFAs of bacterial (i16:0, a16:0, i15:0, a15:0, 16:1ω7,
18:1ω7) and fungal communities (18:2ω6,9; 18:3ω6,9,12; 16:1ω5) are used to draw
conclusions about the qualitative composition of living microbial communities, their
contribution to utilisation of C by various origin (plant or microbial) and to understand
trophic interactions within the soil (Ruess et al., 2005). In contrast, amino sugars
(glucosamine, galactosamine, mannosamine and muramic acid) are usually used to assess the
contributions of bacterial and fungal residues to SOM (Engelking et al., 2007; Glaser et al.,
2004). Muramic acid is of bacterial origin, whereas glucosamine is derived from both fungal
and bacterial cell walls (Glaser et al., 2004). Galactosamine is more abundant in fungal than
in bacterial cell walls (Engelking et al., 2007; Glaser et al., 2004).

Fate of cell membrane and cell wall biomarkers in soil is strongly linked to the

turnover of microorganisms. The cellular turnover of the soil bacterial community is higher
(ca. 2–3-fold per year) than that of the fungal community (ca. 0.75 times per year) (Moore et
al., 2005; Rousk and Baath, 2007; Waring et al., 2013). However, the relationship between
cellular turnover and intracellular C turnover – the question of ecological relevance for the C
cycle – has rarely been investigated. Therefore, if PLFAs characterize the living microbial
community and are rapidly decomposed after cell death (Kindler et al., 2009), a similar
degradation can be assumed for PLFAs molecules originating from various microorganisms.
In contrast, amino sugar polymers display markedly different decomposition kinetics, in that



they can be stabilized in SOM as polymers (Glaser et al., 2004). Thus, the comparison of C
turnover for cell membrane and cell wall components can be used to characterize the
contribution of various microbial groups to short-term C utilisation and to the stabilization of
microbially derived C in SOM.

Combination of PLFAs and amino-sugar biomarkers analyses, as well as cytosolic C

measurement with isotope tracing techniques (based on $^{13}$C natural abundance or $^{13}$C/$^{14}$C
labelling) have been used in various studies to characterize organic C utilisation by the
microbial community (Bai et al., 2013; Brant et al., 2006). However, to date no systematic
studies have compared these contrasting cell compartments in a single soil within a C
turnover experiment. Therefore, this study aimed to examine C allocation to various cell
compartments following $^{13}$C labelling with a ubiquitous monomer, glucose. Glucose has a
higher concentrations in the soil solution compared to other low molecular weight organics
(Fischer et al., 2007), due to its diverse origin: from cellulose decomposition, presence in
rhizodeposition (Derrien et al., 2004; Gunina and Kuzyakov, 2015), and synthesis by
microorganisms. It is also used by most of the microbial groups, and, thus, is the most
suitable substance for such a study.

We analyzed glucose derived $^{13}$C partitioning into the cytosol, cell membranes and

cell walls, to evaluate the turnover time of C in each pool, and to assess the contribution of
bacterial and fungal biomass to SOM. We hypothesized that: 1) C from sugars is first
incorporated into the cytosol, and subsequently into structural compartments such as cell
membranes (PLFA) and cell walls (amino sugars). Thus, the turnover times should increase
in the order: cytosol < PLFA < amino sugars; 2) incorporation of $^{13}$C glucose should be faster
and higher for bacterial than for fungal biomarkers, because bacterial biomass has a faster
cell turnover than fungal biomass; 3) due to amino sugars have long turnover times and are





mainly dominated in microbial necromass, all incorporated $^{13}$C can be related only to living
biomass and allow estimate percent of replaced C in amino sugars of living microorganisms.

**2. Material and Methods**
*2.1. Field site and experimental design*
The $^{13}$C labeling field experiment was established at an agricultural field trial in Hohenpölz,
Germany (49°54'N, 11°08'E, at 500 m a.s.l.). Triticale, wheat and barley were cultivated by a
rotation at the chosen site. The soil type was a loamy haplic Luvisol (IUSS Working group
WRB, 2014) and had the following chemical properties in the uppermost 10 cm: total organic
C content 1.5%, C/N 10.7, pH 6.6, clay content 22%, CEC 13 cmol$_C$ kg$^{-1}$. The annual
precipitation is 870 mm and mean annual temperature is +7 °C.

In summer 2010, following harvest of the triticale, columns (diameter 10 cm and

height 13 cm) were installed to a depth of 10 cm. Each column contained 1.5 kg of soil. The
50 mL of uniformly labelled $^{13}$C glucose (99 atom % $^{13}$C) was injected into the columns via a
syringe at five points inside the column to spread the tracer homogeneously. Syringe was
equipped with a special pipe having length 13 cm and perforated along the whole length,
while the end of the pipe was sealed to prevent glucose injection below of the column. Each
column received 93.4 µmol $^{13}$C of tracer (0.06 µmol $^{13}$C g$^{-1}$soil) and similar amounts of non-
labeled glucose were applied to the control columns. The experiment was done in four field
replicates, which were organized in a randomized block design. Labelled and control columns
were present within each block. For the first 10 days of the experiment the rainfall was
excluded by protective shelter, which was then removed and the experiment was run for 50
days in total. After 3, 10 and 50 days, separate soil columns (four columns where $^{13}$C was
applied and four control columns) were destructively sampled.



The soil was removed from the column, weighed and the water content was
determined in a subsample. Soil moisture was determined by drying samples for 24 h at 105
°C and was essentially constant during the experiment, ranging between 21–25%. Each soil
sample was sieved to <2 mm and divided into three parts. One part was stored frozen (-20ºC)
for PLFA analysis, another was cooled (+5°C) (during one week) before the microbial
biomass analysis, and the rest was freeze-dried and used for amino-sugar analysis and for
measurement of the total amount of glucose derived $^{13}$C remaining in the soil.

*2.2. Bulk soil $\delta^{13}$C analysis*
The soil for the $\delta^{13}$C analysis was milled and $\delta^{13}$C values of bulk SOM were determined
using a Euro EA Elemental Analyser (Eurovector, Milan, Italy) unit coupled via a ConFlo III
interface (Thermo-Fischer, Bremen, Germany) to a Delta V Advantage IRMS (Thermo
Fischer, Bremen, Germany). The amount of glucose derived $^{13}$C remaining in the soil was
calculated based on a mixing model (Equations 1 and 2), where the amount of C in the
background sample in Equation 1 was substituted according to Equation 2.
$[C]_{soil} \cdot at\%_{soil} = [C]_{BG} \cdot at\%_{BG} + [C]_{glc} \cdot at\%_{glc}$ \hfill Eq. (1)
$[C]_{soil} = [C]_{BG} + [C]_{glc}$ \hfill Eq. (2)
with:
$[C]_{soil/BG/glc}$        C amount of enriched soil sample / background soil sample /

glucose derived C in soil            $(mol \cdot g_{soil}^{-1})$

$at\%_{soil/BG/glc}$        $^{13}$C in enriched soil sample / background soil sample /

applied glucose                       (at%)




*2.3. Cytosolic C pool*
The cytosolic pool was determined by the fumigation–extraction technique from fresh soil
shortly after sampling, according to Wu et al. (1990) with slight changes. Briefly, 15 g fresh
soil was placed into glass vials, which were exposed to chloroform during 5 days. After
defumigation, the cytosolic C was extracted from the soil with 45 mL 0.05 M $K_2SO_4$. Due to
fumigation–extraction technique allows to obtain not only soluble components, but also cell
organelles and cell particles, we named pool of C in fumigated extracts as cytosol only for
simplification of terminology. Organic C was measured with a high-temperature combustion
TOC-analyser (Analyser multi N/C 2100, Analytik Jena, Germany). The cytosolic pool was
calculated as the difference between organic C in fumigated and unfumigated samples
without correcting for extraction factor. After organic C concentration were measured, the
$K_2SO_4$ extracts were freeze-dried and the $\delta^{13}C$ values of a 30–35 µg subsample were
determined using EA-IRMS (instrumentation identical to soil $\delta^{13}C$ determination). The
amount of glucose derived $^{13}C$ in fumigated and unfumigated samples was calculated
according to the above-mentioned mixing model (Equations 1 and 2). The $^{13}C$ in the
microbial cytosol was calculated from the difference in these incorporations.

*2.4. Phospholipid fatty acid analysis*
The PLFA analysis was performed using the liquid–liquid extraction method of Frostegard et
al. (1991) with some modifications (Gunina et al., 2014). Briefly, 6 g soil were extracted with
a 25-mL one-phase mixture of chloroform, methanol and 0.15 M aqueous citric acid (1:2:0.8
v/v/v) with two extraction steps. The 19:0-phospholipid (dinonadecanoylglycerol-
phosphatidylcholine, Larodan Lipids, Malmö, Sweden) was used as internal standard one
(IS1) and was added directly to soil before extraction (25 µL with 1 µg µL$^{-1}$). Additional





chloroform and citric acid was added to the extract achieve a separation of two liquid phases,
in which the lipid fraction was separated from other organics. Phospholipids were separated
from neutral- and glycolipids by soild-phase extraction using a silica column. Alkaline
saponification of the purified phospholipids was performed with 0.5 mL 0.5 M NaOH
dissolved in dried MeOH, followed by methylation with 0.75 mL $BF_3$ in methanol. The
resulting fatty acid methyl esters (FAMEs) were purified by liquid–liquid extraction with
hexane (three times). Before the final quality and quantity measurements, internal standard
two (IS2) (13:0 FAME) (15 µL with 1 µg µL$^{-1}$) was added to the samples (Knapp, 1979).

All PLFA samples were analysed by gas chromatograph (GC) (Hewlett Packard 5890

GC coupled to a mass-selective detector 5971A) (Gunina et al., 2014). A 25 m HP-1
methylpolysiloxane column (internal diameter 0.25 mm, film thickness 0.25 µm) was used
(Gunina et al., 2014). Peaks were integrated and the ratio to IS2 was calculated for each peak
per chromatogram. Substances were quantified using a calibration curve, which was
constructed using 29 single standard substances, (13:0, 14:0, i14:0, a14:0, 14:1ω5, 15:0,
i15:0, a15:0; 16:0, a16:0, i16:0, 16:1ω5; 16:1ω7, 10Me16:0, 17:0, a17:0, i17:0, cy17:0, 18:0,
10Me18:0, 18:1ω7, 18:1ω9, 18:2ω6,9, 18:3ω6,9,12, cy19:0, 19:0, 20:0, 20:1ω9, 20:4ω6) at
six concentrations. The recovery of extracted PLFA was calculated using IS1 and the PLFA
contents of samples were individually corrected for recovery. Based on the measured PLFAs
contents, the PLFAs C was calculated for the each single compound.

The $^{13}C/^{12}C$ isotope ratios of the single fatty acids were determined by an IRMS Delta

PlusTM coupled to a gas chromatograph (GC; Trace GC 2000) via a GC-II/III-combustion
interface (all units from Thermo-Fisher, Bremen, Germany) (Gunina et al., 2014). A 15 m
HP-1 methylpolysiloxane column coupled with a 30 m HP-5 (5% Phenyl)-
methylpolysiloxane column (both with an internal diameter of 0.25 mm and a film thickness
of 0.25 µm) were used. The measured δ$^{13}$C values of the fatty acids were corrected for the



225 effect of derivative C by analogy to Glaser and Amelung (2002) and were referenced to Pee

226 Dee Belemnite by external standards. The enrichment of $^{13}$C in single fatty acids was

227 calculated by analogy to bulk soil and cytosol according to Equations 1 and 2, following a

228 two-pool dilution model (Gearing et al., 1991).

230 *2.5. Amino sugar analysis*

231 Acid hydrolysis was performed to obtain amino sugars from soil and further ion removal was

232 performed according to the method of Zhang and Amelung (1996) with optimization for $\delta^{13}$C

233 determination (Glaser and Gross, 2005). Methylglucamine (100 μL, 5 mg mL$^{-1}$) was used as

234 IS1 and was added to the samples after hydrolysis. Following iron and salt removal, non-

235 cationic compounds such as monosaccharides and carboxylic acids were removed from the

236 extracts using a cation exchange column (AG 50W-X8 Resin, H$^{+}$ form, mesh size 100–200,

237 Biorad, Munich, Germany) (Indorf et al., 2012). For final measurement, IS2 – fructose (50

238 μL, 1 mg mL$^{-1}$) – was added to each sample. The amino sugar content and $^{13}$C enrichment

239 were determined by LC-O-IRMS (ICS-5000 SP ion chromatography system coupled by an

240 LC IsoLink to a Delta V Advantage Isotope Ratio Mass Spectrometer (Thermo-Fischer,

241 Bremen, Germany)) (Dippold et al., 2014). Amino sugars were quantified using a calibration

242 curve, which was constructed using four single standard substances (glucosamine,

243 galactosamine, mannosamine and muramic acid) as external standards at four different

244 concentrations (Dippold et al., 2014).


246 *2.6. Calculations and statistical analysis*

247 The assignment of fatty acids to distinct microbial groups was performed by factor analysis

248 with the principal component extraction method in combination with databases (Zelles,



1997). This method enables quality separation of microbial groups within the soils (Apostel
et al., 2013; Gunina et al., 2014). The results of the factor analysis are presented in
Supplementary Table 1.

Total incorporation of glucose derived $^{13}$C ($^{13}$C$_{incorp}$) (means $^{13}$C incorporation

represented as % of total applied $^{13}$C) and enrichment ($^{13}$C$_{enrichm}$) (means $^{13}$C incorporation
represented as % of total C pool) of the cytosol, PLFAs and amino sugars was calculated
according to Equations 3 and 4, respectively. The C turnover time in the cell pools was
calculated as $^{1}/_{k}$; the value of k was obtained from Equation 5.
$$^{13}C_{incorp} = \frac{C_{Glc}}{^{13}C_{Applied}} \times 100\% \qquad\qquad \text{Eq. (3)}$$
$$^{13}C_{enrichm} = \frac{C_{Glc}}{^{Total}C_{Pool}} \times 100\% \qquad\qquad \text{Eq. (4)}$$
with
$C_{Glc}$         amount of glucose derived C incorporated into a distinct cell compartment

calculated by equation (1) and (2)         (µmol $^{13}$C per column)

$^{13}C_{Applied}$    amount of applied glucose $^{13}$C         (µmol $^{13}$C per column)
$^{Total}C_{Pool}$    amount of pool C         (µmol C per column)

$$C_{enrichm(t)} = C_{enrichm(0)} \cdot \exp^{-kt} \qquad\qquad \text{Eq. (5)}$$
with
$C_{enrichm\ (t)}$       $^{13}$C enrichment of the compartment,

obtained from Eq. 4 at time t         (%)

$C_{enrichm\ (0)}$       $^{13}$C enrichment of the compartment

obtained from Eq. 4 at time 0         (%)



k                    decomposition rate constant              (% day$^{-1}$)
t                    time                                     (days)

One-way ANOVA was used to estimate the significance of differences in total $^{13}$C

incorporation and enrichment of cytosol, PLFAs and amino sugars. The data always represent
the mean of four replications ± standard error. To calculate the turnover time of C in the
cytosol, PLFA and amino sugar pools, a single exponential model was used (Eq. 5)
(Kuzyakov, 2011; Parton et al., 1987).

**3. Results**
*3.1. Glucose utilisation and its partitioning within microbial biomass pools*
Amino sugars were the largest pool, due to their accumulation in SOM, whereas pools that
mainly characterize living MB showed smaller C contents (Table 1). The cytosolic pool (C
content 210±7.10 for day 3; 195±14.8 for day 10; 198±19.9 mg C kg$^{-1}$ soil for day 50) as well
as nearly all PLFA groups (Suppl. Table 2) remained constant during the experiment.

**[Table 1]**


The cytosolic pool contained the highest amount of $^{13}$C among the investigated

microbial pools (15–25% of applied $^{13}$C), whereas the lowest amount was recovered in amino
sugars (0.8–1.6% of applied $^{13}$C) (Fig. 1). The amount of glucose derived $^{13}$C in the cytosolic
pool decreased over time, with the largest decrease from day 3 to day 10, and then remained
constant for the following month (Fig. 1). The total $^{13}$C incorporation into PLFA was
generally very low and was in the same range as incorporation into amino sugars (Fig. 1).



The $^{13}$C dynamics in PLFA showed no clear trend (high standard error) (Fig. 1). In contrast,
$^{13}$C in amino sugars increased two fold during the 50 day experiment ($p<0.05$).
**[Fig. 1]**

*3.2. Turnover time of C in microbial biomass pools*
To evaluate C turnover in the cytosol, PLFAs and amino sugars, we calculated the
enrichment (% of incorporated $^{13}$C relatively to pool C) of each pool by glucose derived $^{13}$C.
The enrichment was the highest in PLFAs (Fig. 2) and was 5–8 times lower in the cytosolic
pool. The $^{13}$C enrichment in amino sugars was the lowest (Fig. 2). Based on the decrease of
$^{13}$C enrichment over time (Fig. 2), the C turnover in the cytosol and PLFAs was calculated as
151 and 47 days, respectively. The C turnover time in the amino-sugar pool could not be
calculated by this approach because the maximum incorporation had not yet been reached,
and consequently a decomposition function could not be fitted.
**[Fig. 2]**

*3.3 Phospholipid fatty acids*
Fatty acids of bacterial origin dominated over those of fungal origin within the living
microbial community characterized by PLFA composition (Table 1). Gram-negative (G-)
fatty acids were more abundant than gram-positive (G+) ones. Actinomycetes and vesicular
arbuscular mycorrhiza (VAM) fatty acids dominated in the composition of filamentous
microorganisms, and saprotrophic fungi showed a relatively low presence in PLFAs. The
PLFA content of most groups did not change significantly during the experiment, reflecting
steady-state conditions for the microbial community (see supplementary Table 2).



Glucose derived $^{13}$C was incorporated in higher portions into bacterial than into fungal
PLFAs (Fig. 3, top). Remarkably, the $^{13}$C enrichment decreased over time for all bacterial
PLFAs, whereas it increased or remained constant for VAM, fungi and filamentous, bacterial
actinomycetes (Fig. 3, bottom), indicating differences in C turnover in single-celled
organisms compared to filamentous organisms.

**[Fig. 3]**


*3.4. Amino sugars*
The content of amino sugars followed the order: muramic acid < galactosamine <
glucosamine (Table 1). The glucosamine/muramic acid ratio varied between 17 and 55,
whereas the galactosamine/muramic acid ratio ranged between 12 and 19 (Table 1). This
provides evidence that bacterial residues were dominant in the composition of microbial
residues in SOM.
The incorporation of glucose derived $^{13}$C into amino sugars increased in the order:
muramic acid = galactosamine < glucosamine (Fig. 4, top) reflecting partly their pool sizes.
The $^{13}$C incorporation showed no increase from day 3 to day 50 for any amino sugars. The
ratios of glucosamine/muramic acid and galactosamine/muramic acid, calculated for the
incorporated $^{13}$C, were about six. This is much lower than the ratio observed for the pools of
amino sugars. The $^{13}$C enrichment did not increase from day 3 to day 50 for any of the amino
sugars. The highest enrichment was observed for muramic acid and the lowest for
galactosamine (Fig. 4, bottom). The $^{13}$C enrichment in amino sugars was 10–20 times lower
than for PLFA.

**[Fig. 4]**




**4. Discussion**
*4.1. Glucose decomposition*
The amount of glucose derived $^{13}$C remaining in soil after 50 days was in the range 80 %
which was higher than reported by other studies. Glanville et al. (2012) observed that 50%
of glucose C remained in SOM after 20 days; Wu et al. (1993) reported that 55% of glucose
derived $^{14}$C remained after 50 days; Perelo and Munch (2005) reported the mineralisation of
50% of $^{13}$C glucose within 98 days. The amounts of applied C, as well as differences in
microbial activity in the investigated soils, explain the variation between studies in the
portion of remaining glucose C. The rather high amount of remaining glucose $^{13}$C observed in
this study agrees with results obtained by adding less than 150 µg glucose C g$^{-1}$ soil (2 µmol
C g$^{-1}$ soil) compared to the application of glucose at high addition rates, i.e. more than 150 µg
C g$^{-1}$ soil (Bremer and Kuikman, 1994; Schneckenberger et al., 2008). Glucose C was stored
within the cells due to the starvation conditions of microbial communities, arising from the
general limitation of easily accessible C sources (Bremer and Kuikman, 1994; Schimel and
Weintraub, 2003) due to long term cultivation. This leads to maintenance and starvation
metabolism in microorganisms (Blagodatskaya and Kuzyakov, 2013), where the use of low
molecular weight organic C for energy production, and therefore its mineralisation, are
strongly reduced and conservation of C within the microorganisms prevails (Bremer and
Kuikman, 1994).

The decomposition of 20% of glucose derived $^{13}$C within the first three days (Fig. 1)

agrees with previously reported data (Boddy et al., 2007; Gregorich et al., 1991; Perelo and
Munch, 2005). Glucose is decomposed in soil in two stages (Gunina and Kuzyakov, 2015):
during the first one, part of glucose C is immediately mineralized to $CO_2$ and part is
incorporated into the microbial compartments. This first stage takes place in the first day after
substrate addition and is 30 times faster than the 2$^{nd}$ stage (Gregorich et al., 1991), during





which C incorporated into MB is further transformed and is used for microbial anabolism
e.g., in stable cell polymers, or stored for later catabolism (Bremer and Kuikman, 1994). A
significant portion of glucose derived C was stored in the non-specific pool in SOM (Fig. 1),
e.g., as microbial storage compounds and other cellular building blocks, which can contribute
to C accumulation in microbial residues (Wagner GH, 1968; Zelles et al., 1997; Lutzow et al.,
2006). This part cannot be extracted by the methods applied in this study. The amino sugar
method detects only the peptidoglycan and chitin proportions of the cell walls, whereas other
constituents can not be determined by this method (Glaser et al., 2004). Chloroform
fumigation only partially extracts the cytosolic cell compounds, as high molecular weight
compounds, which interact with the soil matrix, cannot be extracted with low molarity salt
solution.

*4.2. Partitioning of $^{13}$C between cell compounds*
To estimate the residual amount of C derived from applied $^{13}$C-labelled low molecular weight
organic substances (LMWOS), the $^{13}$C in SOM or in the total MB pool is frequently
determined. This approach, however, does not allow the portions of $^{13}$C incorporated into
stable and non-stable C pools to be estimated, because the $^{13}$C in SOM includes the sum of
$^{13}$C in living biomass and $^{13}$C in microbial residues. Furthermore, the living MB contains cell
compartments with a broad spectrum of C turnover times. The approach applied in the
present study allows the partitioning of glucose derived C in living MB to be estimated, as
well as the contribution of LMWOS-C to SOM composition.
*4.3. Cytosol*
We calculated the $^{13}$C enrichment of the cytosolic microbial C pool, extracted after
chloroform fumigation. The estimated turnover time of C in this pool was about 151 days





(Fig. 5). This value lies close to the previously reported range of 87–113 days, for the same
pool for soils incubated for 98 days with $^{13}$C glucose (Perelo and Munch, 2005), but were
lower than MB C turnover time calculated using a conversion factor (2.22) - 82 days, for soils
incubated for 60 days with $^{14}$C glucose (Kouno et al., 2002). About 23% of the remaining
glucose derived $^{13}$C was still present in the cytosol after 50 days (Fig. 1), which is within the
range found in previous studies (Grant et al., 1993). This agrees with the model of Nguyen
and Guckert (2001) for the incorporation of glucose to the cytosol when applied at low input
concentrations and its slow utilisation within the microbial cells.

*4.4. Phospholipid fatty acids*
*4.4.1. Phospholipid fatty acid content and turnover*
Phospholipid fatty acid C comprised 0.27% of the soil organic carbon (SOC). The $^{13}$C
incorporation into PLFAs, in case of constant PLFAs content during the experiment, reflects
microbial activity under steady-state conditions (growth and death of microorganisms occur
with the same rates) and processes of the exchange and replacement of existing PLFAs within
living cells.
Few studies have estimated the C turnover time in PLFAs or the turnover time of
PLFAs themselves in soil as very few options exist to estimate these parameters under
steady-state conditions. The turnover time of $^{13}$C-labelled PLFAs contained in dead microbial
cells, was 2.7 days (Kindler et al., 2009). The PLFAs turnover times estimated in the field
conditions using a $C_3/C_4$ vegetation change (Amelung et al., 2008; Glaser, 2005) or $^{14}$C
dating (Rethemeyer et al., 2005) were between 1 and 80 years. However, these approaches
estimate the turnover time of C bound in PLFA, which can be much older than the PLFA
molecules due to repeated C recycling before incorporation. In contrast, $^{13}$C pulse labeling is





an approach that enables direct estimation of the turnover of freshly added C by the initial
incorporation peak. The approach used in the present study showed that the C turnover time
in PLFA is about 47 days (Fig. 2 and Fig. 5). Accordingly, if the decomposition after cell
death is about three days, the PLFA turnover time in living cells is about 44 days. This short
turnover time of PLFAs is significantly lower than the C turnover time in the cytosol (Fig. 2).
This is because the membrane is the interacting surface between the cell and the environment
and thus, frequent and rapid adaptations of its structure are crucial for active microorganisms.
In contrast, the extracted cytosolic pool includes C from both active and dormant
microorganisms (Blagodatskaya and Kuzyakov, 2013), and the latter can dilute the $^{13}$C signal
incorporated into the active pool with non-labelled C, yielding a lower turnover of this pool.

*4.4.2. Contribution of microbial groups to glucose derived C utilisation*
Bacterial fatty acids dominated in the community structure measured by PLFA, and among
bacteria, fatty acids from G- bacteria were the most abundant (Table 1). Because we used
agricultural soil with pH close to neutral (6.6), the predominance of bacterial PLFAs was
expected. Filamentous microorganisms were represented mainly by actinomycetes and VAM,
which are also typical for agricultural soils (Dungait et al., 2011; McCarthy and Williams,
1992). We classified 16:1ω5 fatty acid as a biomarker for VAM (Olsson, 1999) and not for
G- bacteria, because: i) VAM are usually abundant in soils, where they form a symbiotic
relationship with up to 80% of land plants (Madan et al., 2002), and ii) 16:1ω5 behaved
similarly to fungi in terms of glucose derived C use (Fig. 3, top). For the precise
interpretation of 16:1ω5 as a VAM fatty acid, the simultaneous analysis of 16:1ω5 in
comparison to neutral lipids should be performed, otherwise, the relationship of 16:1ω5 to
VAM should be viewed with caution.





More glucose derived $^{13}$C was incorporated into bacterial PLFAs (Fig. 3, top), than
filamentous microorganisms. This can be a consequence of low C loading rates (less than 4
mg C g-1 soil, see (Reischke et al., 2014), under which conditions the added C is utilized
primarily by bacterial communities, whereas at higher concentrations of applied substrate, the
dominance of fungi in substrate utilisation is observed (Reischke et al., 2014).
Total $^{13}$C incorporation into gram-negative fatty acids was higher (taking both G-
groups together) compared to G+ bacterial PLFAs (Fig. 3, top), which might be due to: i) the
abundance of their fatty acids, which was higher (Table 1) or ii) glucose uptake activity,
which was higher for G- than G+ groups. In contrast, the $^{13}$C enrichment (total $^{13}$C
incorporation related to C in particular biomarkers) for G- bacterial PLFAs was not higher
than that for G+ (Fig. 3, bottom). Thus, the high total $^{13}$C incorporation into G- bacterial
biomarkers can mainly corresponds to their high content in the soil, not to higher activity of
microbial groups. However, replacement of PLFAs C by glucose derived $^{13}$C is only a proxy
of microbial activity and only partly confirm the real mechanisms. This clearly suggests that
the analysis of isotope data after labeling in general requires the calculation and combined
interpretation of both the total tracer C incorporation as well as the $^{13}$C enrichment in the
investigated pool.
In contrast to our results, a higher incorporation of glucose derived $^{13}$C into G+ than
G- PLFAs was observed in other studies (Dungait et al., 2011; Ziegler et al., 2005). However,
in these studies, much higher amounts of C were applied to the soil (15 μg C g$^{-1}$ soil), which
stimulated the growth of G+ bacteria. In contrast, under steady-state conditions with low
glucose concentrations in soil, G- bacteria were the most competitive group for glucose
uptake (Fig. 3).
The $^{13}$C enrichment of bacterial PLFAs decreased from day 3 to day 50, whereas $^{13}$C
in fungal PLFAs increased (in the case of VAM) or stayed constant (Fig. 3, bottom). The



decrease in $^{13}$C enrichment in bacterial fatty acids indicates a partial turnover of bacterial
lipid membranes, which is much faster than turnover in fungal membranes. This result is
consistent with the turnover time of bacterial biomass in soil (Baath, 1998), which is about 10
days, whereas fungal biomass turnover times range between 130–150 days (Rousk and Baath,
2007). Consequently, the increase in $^{13}$C enrichment in fungal PLFAs at late sampling points
indicates that fungi consume the exudation products of bacteria or even dead bacterial
biomass (Zhang et al., 2013; Ziegler et al., 2005).

*4.5. Amino sugars*
*4.5.1. Amino sugar content and amino sugar C turnover in total and living microbial cell*
*walls*
Amino sugars represented the largest microbial pool investigated in this study (Table 1) and
comprised 3.7% of SOC. Chitin and peptidoglycan, the direct sources of the amino sugars,
comprise no more than 5% of cell biomass (Park and Uehara, 2008; Wallander et al., 2013).
Therefore, the high amount of amino sugars relative to PLFA can only be explained by their
high proportion in microbial residues/necromass (Glaser et al., 2004; Liang et al., 2008;
Glaser et al., 2004). Irrespective of the large pool size of the amino sugars, their incorporation
of glucose derived $^{13}$C was the lowest compared to other compartments in living cells. The
total $^{13}$C incorporation and enrichment of amino sugars increased from day 3 to day 50, in
contrast to cytosol and PLFA pools. Consequently, amino sugars have the slowest turnover in
soils, presumably even within living cells, for two reasons: 1) cell walls are polymers that
require a rather complex biosynthesis of the amino-sugar fibers, 2) cell-wall polymerization
occurs extracellularly (Lengeler et al., 1999) and 3) microorganisms do not need to



synthesize peptidoglycan unless they multiply. To calculate C turnover time in this pool,
further long-term sampling points with $^{13}$C-amino-sugar analysis are necessary.

The majority of amino sugars extracted after acid hydrolysis represent microbial

necromass, which does not incorporate any glucose derived $^{13}$C, but strongly dilutes the $^{13}$C
incorporated into the walls of living cells. To estimate the $^{13}$C incorporation into amino
sugars of living cells, we first calculated the amount of amino sugars in the living MB pool
based on the fatty acids content. Assuming that PLFAs are present only in living biomass,
and that the ratio of fatty acids to amino sugars in living biomass is about 0.23 (Lengeler et
al., 1999), we estimated the amount of amino sugars in living MB to be 0.20 µmol g$^{-1}$ soil
fatty acids/0.23 = 0.87 µmol g$^{-1}$ soil. The estimated percentage of amino sugars in living
biomass from the total amino sugar pool was 0.87/7.70 (total AS (µmol g$^{-1}$ soil))*100 = 11%.
This estimate agrees with that of Amelung et al. (2001a) and Glaser et al. (2004), who
reported that the amount of amino sugars in living biomass is one to two orders of magnitude
lower than in the total amino-sugar pool. We calculated the $^{13}$C enrichment in amino sugars
for the first sampling point, assuming that all replaced C is still contained within living MB
after three days of glucose C utilisation. Total tracer incorporation into amino sugars
consisted of 0.00071 µmol glucose derived $^{13}$C in amino sugars g$^{-1}$ soil/0.87 (µmol amino
sugars g$^{-1}$ soil)*7 (mean amount of C atoms in amino sugars)*100 = 0.57% of the C pool.
Comparison of these data with the $^{13}$C enrichment into PLFAs and the cytosol allowed us to
conclude that the replacement of the amino sugar C with glucose derived $^{13}$C in living
biomass is two-fold slower than the replacement in PLFAs, and faster than in the cytosolic
pool. This reflects that microbial C turnover is a phenomenon that is not restricted to the
death or growth of new cells, but that even within living cells, highly polymeric cell
compounds, including cell walls, are constantly replaced and renewed (Park and Uehara,

2008).



### 4.5.2. Contribution of bacterial and fungal cell walls to SOC


Glucosamine was the dominant amino sugar in the soil, whereas muramic acid was the least
abundant (Table 1), which agrees with most literature data (Engelking et al., 2007; Glaser et
al., 2004). To conclude about the proportions of bacterial and fungal residues in the SOM, the
ratios of glucosamine/muramic acid and galactosamine/muramic acid (Glaser et al., 2004)
were calculated (Table 1). Based on the galactosamine/muramic acid ratio, bacteria were
dominant within the soil microbial community, whereas the glucosamine/muramic acid ratio
indicated that the fungal contribution was larger. This discrepancy is due to unknown sources
of glucosamine in the soil (Glaser et al., 2004), i.e. it originates from bacterial (Amelung et
al., 2001b) and fungal cell walls (Fernandez and Koide, 2012; Glaser et al., 2004; Park and
Uehara, 2008) as well as from actinomycetes, insect and invertebrate. Moreover, previous
studies have confirmed galactosamine to be the most characteristic amino sugar for fungal
residues in soil (Engelking et al., 2007) and thus, galactosamine/muramic acid ratio is more
useful for estimation the composition of microbial residues in SOM. The bacterial origin of
microbial residues in the soil is supported by: 1) the dominance of bacterial PLFA biomarkers
2) the environmental conditions of our the site, namely, long-term agricultural use, which
promotes the dominance of bacterial communities.
Three-fold more glucose derived $^{13}$C was incorporated into glucosamine than into
galactosamine and muramic acid (Fig. 4, top). This correlates with the pool size and indicates
that glucosamine is the most dominant amino sugar not only in total amino sugars, but also
within the walls of living cells. The galactosamine/muramic acid ratio of the incorporated $^{13}$C
was six, and consequently was significantly lower than the ratio calculated for the amount of
amino sugars (Table 1). This indicates that bacteria are more active in glucose derived $^{13}$C
utilisation than fungi, a conclusion also supported by the $^{13}$C-PLFA data (Fig. 3). Thus, even
if the composition of amino sugars does not allow a clear conclusion concerning living



microbial communities in soil, amino sugar analysis combined with $^{13}$C labeling reveals the
activity of living microbial groups in terms of substrate utilisation.

The calculated $^{13}$C enrichment was the highest in muramic acid (Fig. 4, bottom). This

is in agreement with the high $^{13}$C enrichment of bacterial PLFAs compared to VAM and
fungi (Fig. 3). Due to differences in cell-wall architecture, G+ bacteria contain more muramic
acid (approximately four times) than G- bacteria (Lengeler et al., 1999), and thus make a
higher contribution to the $^{13}$C enrichment of muramic acid.

The $^{13}$C enrichment of glucosamine was two-fold lower than muramic acid (Fig. 4,

bottom). This confirms the hypothesis that glucosamine originates from bacterial as well as
fungal cell walls and consequently has a mixed enrichment between that of the fungal
galactosamine and the bacterial muramic acid.

**5. Conclusions**
Tracing the $^{13}$C labelled glucose through cytosol, PLFAs and amino sugars is a prerequisite
for understanding the fate of organic substrates in soil. The highest $^{13}$C enrichment, and thus
turnover of C, was found for the PLFA pool, corresponding to a turnover time of 47 days,
whereas the turnover was slower in the cytosol (150 days). These differences can be
attributed to 1) efficient C recycling in the cytosol, and 2) its heterogeneous composition,
which contains compounds with different turnover rates. The $^{13}$C enrichment of amino sugars
was still increasing at the end of the experiment, reflecting the slowest C turnover within the
investigated pools and that most of this pool consisted of microbial residues and not living
biomass. An approximate calculation of $^{13}$C enrichment of amino sugars in the living biomass
gave values 0.57%  of pool size, which was still lower than for PLFAs. Thus, C turnover in



membrane components is faster than in cell wall components, even if only the portion of the
amino sugar pool in living biomass is considered.

Bacterial PLFAs dominated in the microbial community composition, and much

higher glucose derived $^{13}$C was incorporated to bacterial than to fungal PLFAs too. This
agrees with prevailing role of bacteria in the utilisation of easily available organic substrates
that are present at low concentrations in soil. A slower turnover of filamentous and mainly
fungal biomass compared to bacteria was observed in the $^{13}$C enrichment of the respective
PLFAs. Therefore, filamentous organisms might utilize the products of bacterial metabolism
and biomass, which is an important link in the soil food web.

The galactosamine/muramic acid ratio was between 12 and 19, indicating a

predominance of bacterial vs. fungal residues in SOM. The ratio of galactosamine/muramic
acid for incorporated $^{13}$C confirmed that bacteria were more active in glucose utilisation than
fungi. The $^{13}$C enrichment was the highest for muramic acid and the lowest for
galactosamine, demonstrating that the turnover of bacterial cell wall components is more
rapid than fungal.

Consequently, the combination of $^{13}$C labeling with the subsequent analysis of several

microbial cell compartments and biomarkers is a unique approach to understanding C
partitioning within microbial cells and the microbial communies in soil. This knowledge is
not only crucial for assessing C fluxes and recycling in soil, but is also of special importance
concerning the contribution of C from microbial residues to SOM.

**Author contribution**
Y. Kuzyakov and B. Glaser designed the experiments and M. Dippold and A. Gunina carried
them out. A. Gunina prepared the manuscript with contributions from all co-authors.





**Data availability**

Underlying research data can be accessed by a request from the first author of paper.

**Acknowledgements**

This study was supported by a grant from the Deutsche Forschungsgemeinschaft (DFG KU 1184 19/1 and INST 186/1006-1 FUGG). The authors are grateful to Stefanie Bösel, a technical staff member of the Department of Soil Biochemistry, Institute of Agricultural and Nutritional Science, Martin-Luther University Halle-Wittenberg for performing the bulk isotope and amino sugars measurements. Thanks are extended to MolTer and DAAD, which provided a fellowship for A. Gunina. We are very grateful to the Centre for Stable Isotope Research and Analysis (KOSI) of Göttingen University for the $\delta^{13}C$ measurements.



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





**Table 1** Amount of microbial biomass compartments, their C content, PLFA content of
microbial groups and composition of microbial residues in investigated soil. G-1 and G-2 are
gram-negative group one and two, respectively; G+1 and G+2 are gram positive group one
and two, respectively; Ac – actinomycetes; VAM - vesicular arbuscular mycorrhiza fungi.
Data present mean of three time points ± SE

| Compartment | mg component C kg$^{-1}$ soil | mg kg$^{-1}$ soil | Ratio |
|---|---|---|---|
| **Cytosol** | 201.0 ± 7.1 | - | |
| **Phospholipid fatty acids** | 39.4 ± 4.7 | 51.9 ± 6.2 | |
| **Specific phospholipid fatty acids** | | | |
| G-1 | 8.9±3.6 | 11.6 ± 4.6 | |
| G-2 | 5.6±0.8 | 7.4 ± 1.1 | |
| G+1 | 5.9±1.2 | 7.9 ± 1.6 | |
| G+2 | 0.7±0.3 | 1.0 ± 0.4 | |
| Ac | 2.3±0.7 | 3.0 ± 1.0 | |
| VAM | 1.7±0.3 | 2.2 ± 0.3 | |
| Fungi | 1.0±0.2 | 1.3 ± 0.2 | |
| Bacteria/Fungi | | | 6 - 8.5 |
| **Amino sugars** | 560.7 ± 68.2 | 1393.8 ± 170.0 | |
| Glucosamine | 460.7±79.3 | 1146.5 ± 197.3 | |
| Galactosamine | 90.9±11.3 | 226.3 ± 28.2 | |
| Muramic acid | 9.1±1.8 | 21.1 ± 4.1 | |
| Glucosamine/muramic acid | | | 17 - 55 |
| Glucosamine/muramic acid (literature data for pure cultures*) | | Bacteria | 5.3 |
| | | Fungi | 271 |
| Galactosamine/muramic acid | | | 12 - 19 |
| Galactosamine/muramic acid literature data for pure cultures*) | | Bacteria | 2.8 |
| | | Fungi | 59 |

*Data are taken from Glaser et al. (2004).









**Table captions**
**Table 1** Amount of microbial biomass compartments, their C content, content of microbial
groups and composition of microbial residues in investigated soil. G-1 and G-2 are gram-
negative group one and two, respectively; G+1 and G+2 are gram positive group one and
two, respectively; Ac – actinomycetes; VAM - vesicular arbuscular mycorrhiza fungi.

**Figure captions**
**Fig. 01** Partitioning of glucose derived $^{13}$C in SOM presented as the total $^{13}$C incorporation
between the following pools: non-specified SOM, cytosolic, PLFAs and amino sugars. Small
letters reflect differences between the sampling points for the distinct pool. Data present
mean (n=4) and bars present standard errors.

**Fig. 02** $^{13}$C enrichment in the cytosolic, PLFA and amino-sugar cell pools as well as
functions to calculate the C turnover times in these microbial cell pools. The left y-axis
represents the PLFA pool, the first right y-axis, the cytosolic and the second y-axis, the
amino-sugar pool. Data present mean (n=4) and bars present standard errors.

**Fig. 03** Total incorporation of glucose derived $^{13}$C (top) and $^{13}$C enrichment (bottom) of the
microbial PLFAs. Note that the values for VAM and fungi are scaled-up 10 times (secondary
Y axis) compared to those of other groups (Y axis at the left). Data present mean (n=4) and
bars present standard errors. Small letters reflect differences between the microbial groups for
total $^{13}$C incorporation and $^{13}$C enrichment from glucose; letters a-d are for day three, l-o are
for day 10, x-z are for day 50.



**Fig. 04** Total incorporation of glucose derived $^{13}$C (top) and $^{13}$C enrichment (bottom) of
amino sugars and muramic acid. Letters reflect significant differences in the total
incorporation and $^{13}$C enrichment from glucose into amino sugars on a particular day; letters
a-b are for day three, l-m are for day 10, x-y are for day 50. No significant differences were
observed between the three sampling days. Data present mean (n=4) and bars present
standard errors.
**Fig. 05** Dynamic relationship of microbial utilization of glucose and turnover of cytosol, cell
membrane and cell wall components.



Figure 01.

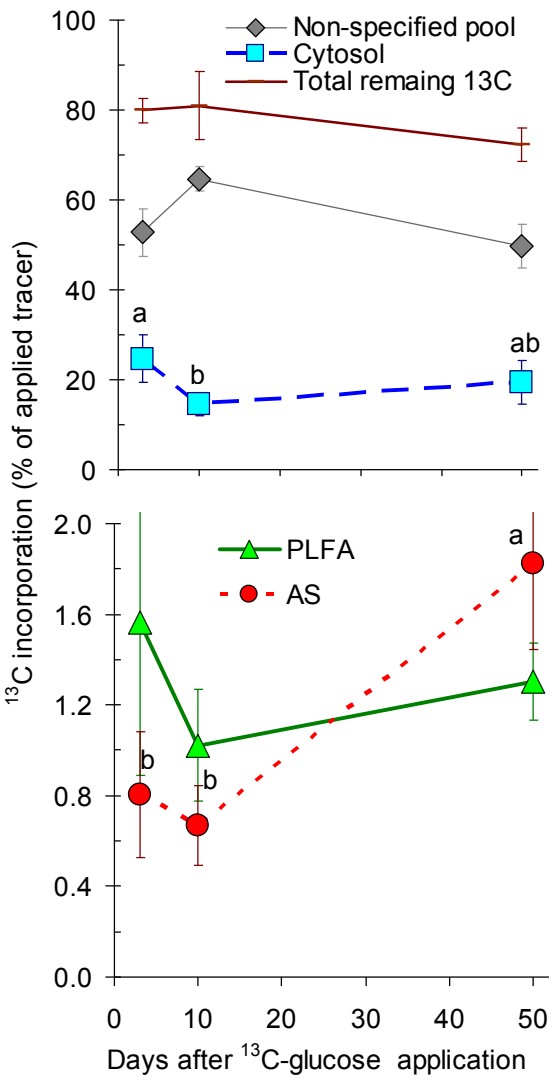





Figure 02.

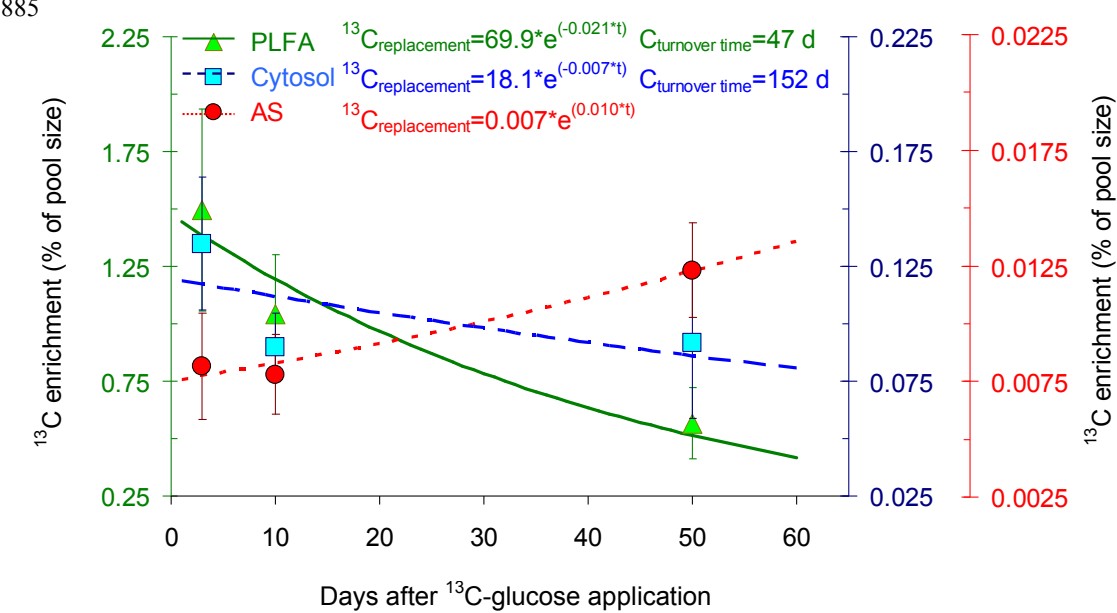





Figure 03.

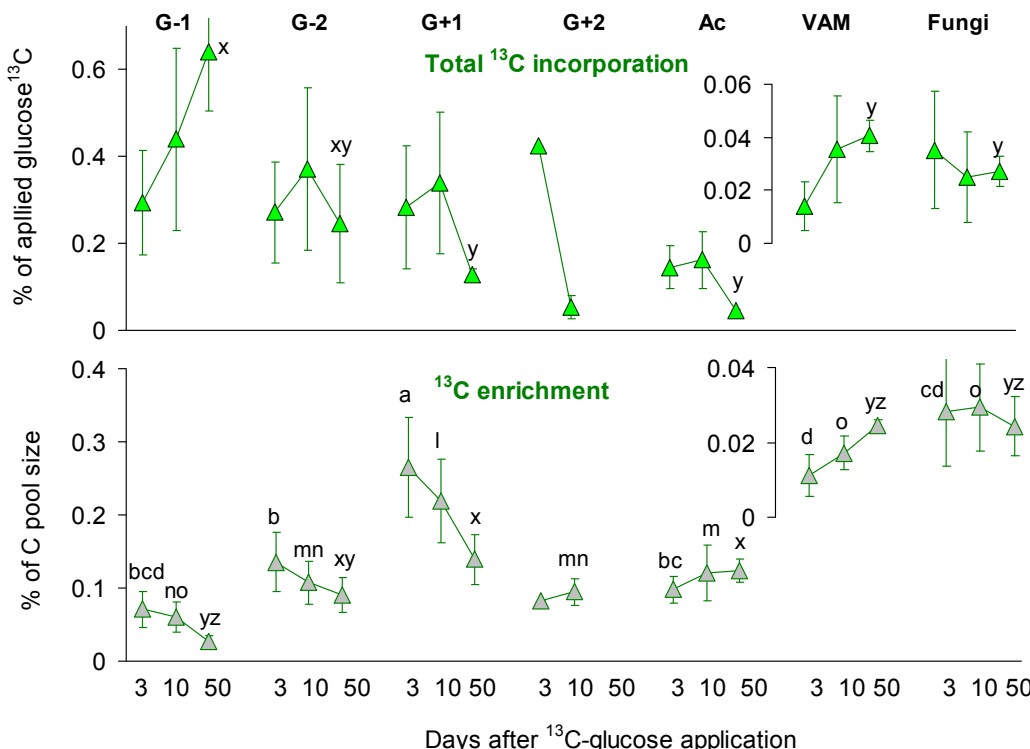



Figure 04.

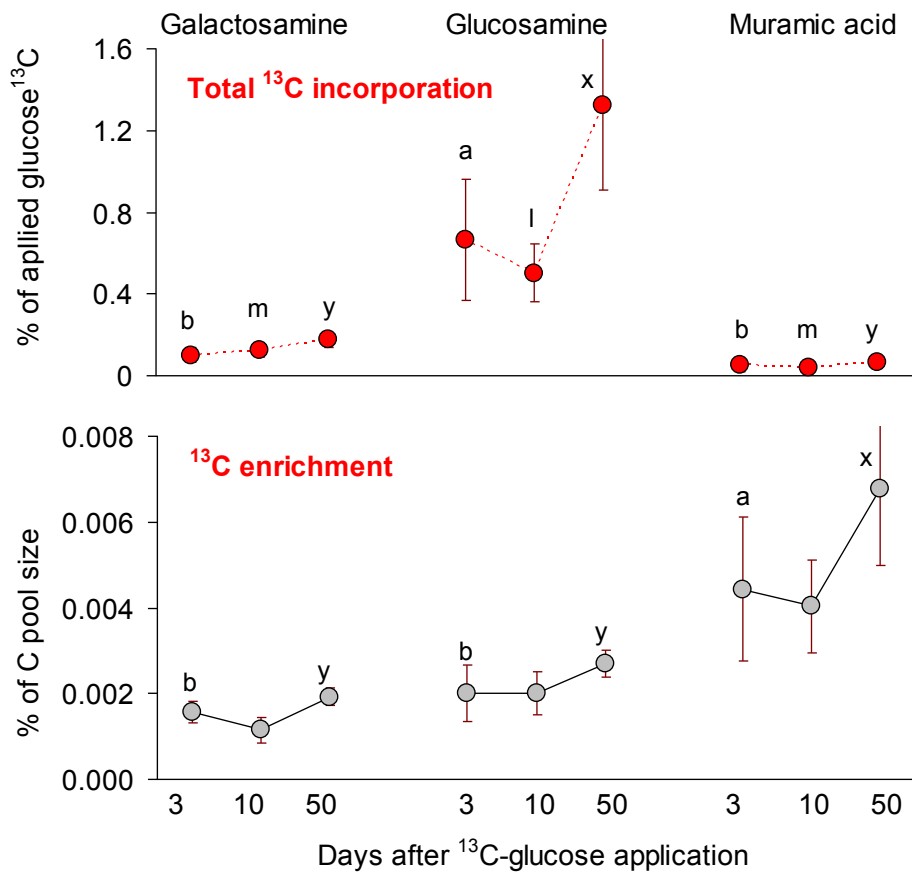




Figure 05.

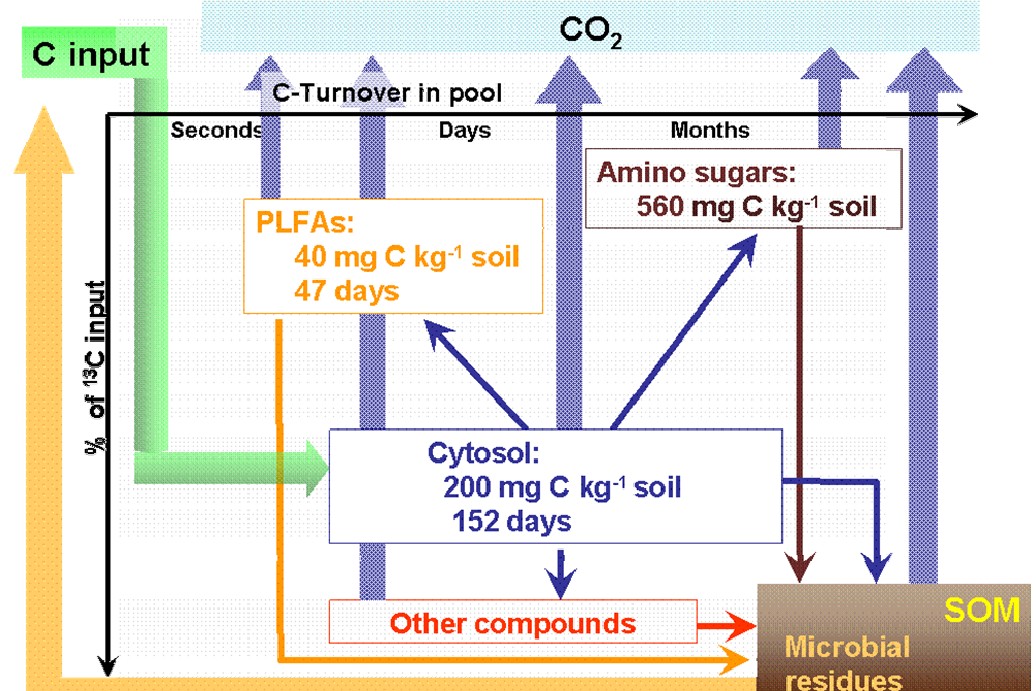



