# Peer review of "Carbon turnover in cell compartments and microbial groups in soil"

_Biogeosciences, 2016_

## Short Comment (SC1) · 28 Jun 2016

I have revised ms CARBON TURNOVER IN CELL COMPARTMENTS AND MICRO-BIAL GROUPS IN SOIL by Gunina et al. To reveal the contribution of particular microbial groups to C utilization and C turnover within the microbial cells, the fate of 13C-labeled glucose was studied under field conditions in this work. The 13C was traced in cytosolic substances, amino sugars and phospholipid fatty acids (PLFA) at intervals of 3, 10 and 50 days after glucose addition. The strong and positive site of this manuscript is the combination of 13C labeling with the subsequent analysis of several microbial cell compartments and biomarkers as a unique approach to understanding C partitioning within microbial cells and the microbial communities in soil. This knowledge is crucial for both assessing C fluxes and their recycling in soil, and studying the contribution of C from microbial residues to SOM. There are some weaknesses of the manuscript, but

they do not affect too much on the paper quality. Below I have included general and specific comments regarding all sections, and after all I recommend to authors revise, clarify and substitute some information through the whole text. To conclude, I think present manuscript may be published in Biogeoscience journal.

General comments

Materials and methods: There is no information on bulk density of the soil in the paper. Provide this information, as it is not clear how you calculated soil mass in cylinder Check all links to figures, equations and tables and present them whether in full text or shortened form (fig, eq., etc), but the same through the whole manuscript.

Specific comments

L 180 – 183 I recommend to change "due to "in "Due to fumigation–extraction technique allows. . ..." for "As fumigation–extraction technique allows. . .."

L 186 Revise sentence: In After organic C concentration were measured → After organic C concentrations were measured OR After organic C concentration was measured

L 295 Clarify if it was 50th day of experiment or after 50 days in the sentence "13C in amino sugars increased two fold during the 50 day experiment (p<0.05)"

L 301-302 Paraphrase the sentence, as it is difficult to read: "The enrichment was the highest in PLFAs (Fig. 2) and was 5–8 times lower in the cytosolic pool. The 13C enrichment in amino sugars was the lowest (Fig. 2) "

L 427- 429 Substitute "we used agricultural soil" on "we conduct experiment on agricultural soil". In your paper you wrote about field experiment, not incubation one with agricultural soil. It is not correct "to use" soil in this situation.

L 450-451 Revise sentence "However, replacement of PLFAs C by glucose derived 13C is only a proxy of microbial activity and only partly confirms the real mechanisms."

L 486-487 What long-term sampling points mean? Maybe, you suggest conducting long-term experiment for turnover time calculation? Or add sampling points to existing experiment?

L 526- 527 Substitute "our the site" on "experimental site or site" in "the environmental conditions of our the site, namely, long-term agricultural use. . .."

Tables

L 809 It will be better to write "Data present mean of three replicates" instead of "Data present mean of three time points".

Figures

L 887 It is difficult to read graphs for Ac. Expand the space between Ac and second scale for VAM and Fungi.

---

## Referee Comment (RC1) · P. Dijkstra (Referee) · 11 Jul 2016

Review of manuscript "Carbon turnover in cell compartments and microbial groups in soil" by Gunina et al.

The authors of this manuscript have analyzed the turnover of different cellular compounds/fractions for different microbial groups using a $^{13}$C labeling experiment (3, 10 and 50 days). This is clearly a worthy and important goal. The experiment is done well although the number of harvests (3) is minimal for this determination of turnover. For reasons described below, I think the manuscript is not acceptable for publication in its current form.

The goal of the manuscript is to evaluate the turnover time of C in each pool and to assess the contribution of bacteria and fungi to SOM. A second goal is to determine the turnover time for different categories of microbes. They hypothesize that turnover time is short for cytosol, intermediate for PLFAs, and long for amino-sugars. However, the results they find indicate that turnover time of lipids<aminosugars<cytosol. They hypothesize that, based on aminosugar ratios, the bacteria contribute more to SOM than fungi, however, the results are contradictory (one ratio suggests bacteria, the other fungi). Instead of defending these observations and rejecting the hypotheses, complex reasons are proposed why turnover time of the cytosol is long, but it is still a "labile pool" that turns over fast but has tight cycling, and, in the discussion, it turns out that one of the aminosugar ratios is "better" than the other, so that the bacterial contribution to SOM is high. In other words, experimental results could not cause rejection of the hypotheses, therefore I have to conclude the experiment was poorly designed and not able to test the proposed hypotheses.

There are several reasons for the inability of this experiment to deliver results that are strong enough to test the hypotheses

1) It is unclear what "cytosol" is and why it is thought to be labile (L37). Although aminosugars and PLFAs are (bio)chemically distinct, this is less so for the fraction "cytosol" (L121, L179 and following). In order to understand the differences between lipids, aminosugars and "cytosol", the authors will have to analyze the amount of lipids and aminosugars in the cytosol fraction.

2) The experiment was not long enough to calculate turnover time for aminosugars (Fig. 2; ). Moreover, although turnover is calculated using one exponential declining function (Fig. 2), in the discussion, a whole paragraph is dedicated stating that glucose decomposition is bi-phasic (L 362), and so the use of a single exponential function needs to be defended. Furthermore, conclusions about turnover rates are presented for PFLAs and aminosugars, without numbers to back up the conclusions. This is because of increasing 13C contents with time for aminosugars and fungal PLFAs; however, if the turnover times cannot be calculated, the conclusion should not be drawn, data should not have been presented (under this title) and/ or more data should be collected. Additionally, turnover rates should have been calculated for the various bacterial and fungal groups based on PFLA data (according to the title). Finally, the presented turnover rates are presented without an estimate of the error associated with it (for example R2 value in Fig. 2, 3 and 4, SE for the turnover time values), making it impossible to evaluate whether the estimated turnover times for lipids and cytosol are truly different.

3) Hypothesis 1 is interesting, but cannot be tested in this experiment, as the initial uptake and incorporation in cytosol and other pools is fast. For example, Frey et al 2013 show that glucose uptake and incorporation in microbial "cytosol" occurs within 6 hours. The authors need to

explain why and how this hypothesis can be tested using the experiment they designed. Hypothesis 3 is not a hypothesis but a (simplifying) assumption, used to interpret the results of this study, not a testable hypothesis. Moreover, the assumption is by definition wrong, but at best is an acceptable approximation. However, no evidence is given to support this assumption. Is 50 days incubation still short enough that no aminosugars are transferred to the necromass pool? In general, the hypotheses are poorly defended or explained mechanistically.

Additional general comments

- The statistics need to be further developed. The estimates of the turnover for the different fractions/compounds (L 304) need to be described with a mean and error. R2, significance and SE need to be added with Fig 2, 3 and 4. Current description does not make it possible to verify the assertion of the authors that the turnover rates of the various pools are significantly different. Fig. 5 does not add to understanding or interpretation of the results and can be removed.
- The observation that the 13C incorporation (as a percentage) was higher in PLFA than in cytosol does not logically result in a conclusion that the incorporation is faster (L32). This result may just be a reflection of the size of the pool (PFLA versus "cytosol"), and certainly does not show "the importance" of membranes "for initial C utilization".
- The use of the term filamentous organisms should be avoided. The authors probably mean fungi. I like the intent of L46, however, the comparison of the dynamic behavior of the three pools remains poorly developed.
- Careless use of references: L 68: Malik et al have not reported on cytosol, nor on its supposed heterogeneity. It is not at all clear how location would affect the turnover time of membranes and cell walls (L70). Bremer and Kuikman (1993; L 73) are not experts in microbial physiology, and therefore not an authorative source to support the statement that labeled glucose appears first in the "cytosol". In fact, they only looked at the cytosol (fumigation-extractable) so cannot comment on whether other compounds or fractions become labeled first or later. A reference is needed to support the assumption that "the cytosol is considered to be the most dynamic pool within microbial cells". Furthermore, heterogeneity (L75-76) has never stopped any calculation of turnover times, as is evident in soil organic matter turnover studies. Important references are missing for example those by Malik et al 2015 where comparison between "cytosol" and PFLAs are made (and DNA/RNA).
- L96 and following: This paragraph tries to distinguish between cellular turnover – I assume as a consequence of cell death is what is referred to here – and turnover of compounds within a living cell. However, it is not that easy to make that distinction – how does one distinguish between lipids being recycled and reused, taken apart and made into for example amino-acids, while other amino acids are recycled into lipids, and what happens after cell death – uptake of lipids by other organisms intact incorporated, reused, recycled, taken apart and/or turned into CO2. Moreover, the observations of increasing 13C concentrations for fungi versus decreasing ones in bacteria suggests some transfer of compounds, but remains unexplained in this manuscript.
- L 146: it is not clear to me why unlabeled glucose was added to the control treatments.
- L 149: explain why the shelter was put in place and why it was removed. What was the effect of this on the soil moisture content?

- L 157: why was the soil stored at 5 °C for 5 days prior to chloroform-fumigation analysis? What happened to the "cytosol" during that time? Does this mean that the value for cytosol is really the value after 8, 15, and 55 days?
- L 180: defumigation is not a word.
- L 186: "extraction efficiency" not "extraction factor"
- L 247: "the assignment of fatty acids to microbial groups …" is confusing me. Does this mean that as part of this study, biomarker PFLAs are assigned to group independent of what is done in other studies?
- L 249: this procedure is not clear to me, but I am not at all familiar with PFLA/Microbial community analysis. My first impression was that the analysis is basically a community analysis – showing, based on PFLAs, what the community looks like. However, L 247 suggests that with this procedure, PFLA are assigned to microbial taxa, but then in the heading of Supplementary Table 1 it suggests that literature data is used – Please clarify what the table is used for, how (and what) literature data is used, and what the results of this analysis means for your experiment. Similarly, L 431: the arguments for using the 16:1w5 as a biomarker for VAM and not G- are weak. The abundance of VAM needs to be expressed relative to G- bacteria. Table one suggest that the total C for PFLAs is higher than for VAM, thus is more abundant (?).
- L290: the description of the results (declines between 3 and 10 days but then remains constant then constant) does not match the assumed exponential decline. Please explain.
- Fig. 2, 3, 4: the statistical tests should also be done between harvests, not only between microbial groups.
- L 347: the explanation for the differences between this study and published results, namely the amount of glucose added and the microbial activity, are not revealed. Some further information on these explanatory variables would be appreciated. Is microbial activity measured in this study, microbial activity is not measured? The idea that microbes store glucose when added in small quantities is unproven – it is a mere assumption, recently defended by Sinsabaugh et al 2013, but evidence for storage was absent in recent experiments by Dijkstra et al (2015). The idea that the storage leads to maintenance is in contradiction to the 80% recovery after 50 days, and with the idea that microbial pools and cells turn over fast.
- L 362: the description of the two stages of glucose decomposition – 1) CO2 production plus biosynthesis, and 2) C incorporated in microbial cells is used for anabolism is confusing. Is anabolism different from biosynthesis? Is during the second phase CO2 production absent? How do the two phases relate to the biosynthesis of lipids, cytosol, and aminosugars? Please clarify
- L 395: what is this model, please explain some salient details and how it agrees with your observations.
- L 419: this rationalization needs some references or evidence that contact with the environment leads to rapid turnover.
- L 421: the problem of active and inactive cells for cytosol dynamics is similar for lipid dynamics, as inactive cells also have membranes.
- L 482: how is this conclusion drawn when the turnover rate cannot be calculated according to L 486. L 506: how do you determine that the turnover of the amino-sugars is higher than that of the cytosol pool? L 509: this would be a wonderful conclusion, but it does not appear in the abstract at all. What is the reason that the cytosol is so stable? Please elaborate.

- L 511 and following – the results from the measurements seem to indicate contrasting conclusions – bacteria or fungi are most important (L516 and following). It is then stated that only the galactosamine/muramic acid ratio should be used. So, this means that the reader has wasted a number of valuable brain cells thinking about the galactosamine/glucosamine ratios, and looked at the data, but that was all a waste of time? Why not start with what is known (galac/muramic ratio) and leave it at that. Furthermore, there is a lot more text about the three aminosugars and their ratios in relation to bacteria and fungi – is that still relevant inlight of L 521?
- Fig 1: explain what is total 13C remaining, what is non-specified pool? Remake the Fig so that the SE of the aminosugars are fully shown.
- Fig. 2: what is the equation with the word "replacement" in it? I think it is just the function of $^{13}$C over time, and thus the word replacement can be removed, but I may be wrong. Add $R^2$, P value and significance (and SE of the turnover estimate)
- Fig. 3: instead of showing differences between microbial groups, we need to know the differences between dates AND microbial groups to evaluate how these differences represent significant differences in turnover, and whether this turnover differs between groups. Moreover, the goal of this paper was to determine differences in turnover between microbial groups, but this is not calculated. If turnover cannot be calculated for groups where $^{13}$C enrichment is increasing over time, what was the basis for the conclusion that turnover differed between fungi and bacteria (L320)?
- Fig 5: not really helpful.

Paul Dijkstra

---

## Referee Comment (RC2) · Anonymous Referee #2 · 26 Jul 2016

Contributing to the understanding the C turnover of different cell compartments is certainly a valuable goal. This manuscript reports the findings of a 3-50 day incubation experiment where 13C from glucose is followed into the cytosol, PLFAs and amino sugars. They set out to test three hypotheses: (1) turnover times increase in the order cytosol>PLFA>amino sugars; (2) incorporation of 13C is faster for bacterial than fungal markers; (3) "due to amino sugars have long turnover times and are mainly dominated in microbial necromass, all incorported 13C can be related only to living biomass and allow estimate percent of replaced C in amino sugars of living organisms. I am concerned about the limitations brought by sampling intensity and analysis of the data and mostly that the findings that were possible with this study don't really allow for testing the hypotheses presented. Probably as a result. A lot of what's presented in the Discussion is not critical or essential and it's either general, repetitive or tangential.

[Figure]

1. Hypothesis 1 could not really be tested given the duration and intervals of sampling during the experiment. Glucose gets processed, incorporated, lost and recycled into PLFA already in the first two days (e.g. Ziegler 2005) and this can vary with soil and environmental conditions. Starting measurements after 3 days leaves us without any information of when the peak of uptake took place (thus when time zero for decomposition started) and when recycling started which would matter for trying to estimating turnover. Then on the other end, 50 day was not sufficient time for the aminosugars to finish building up and start decomposing, as the authors discuss. Perhaps the data can be used to answer a different question. From Figure 2, we don't know how good the model fits were (it would be important given there were three points and large error bars).

2. Hypothesis 2 is about differential incorporation by bacteria and fungi (who incorporates it faster). AGain, missing the first three day is pretty critical (Ziegler 2005 clearly shows this). There are already many experiments that have assessed "initial" incorportation of 13C into biomarker lipids and we wouldn't then need 50 days incubation for this. Also, I am surprised that there was no effect of time on the composition of the PLFA as 50 days is quite a long time for microbes and PLFA profiles tend to be more sensitive to time than any other driver, and, C depletion in 50 days of an incubation would be substantial.

3. Hypothesis 3 is hard to follow. I tried but was not able to understand it.

4. In the Introduction, the paragraph comprising Lines 96-108 is a very convoluted and hard to follow, however, it refers to the main rational for carrying out this study.

5. I don't understand why there was not an attempt to estimate ks for the PLFAs. That would be the main purpose of this approach, in my view. Also, how can the VAM be building up 13C, if they are mycorrizhae? This probably suggest the marker was indicating a saprotroph, not mycorrhizae, which is known to happen in soils.

6. About the aminosugars, what we still don't know and doesn't get explored and

discussed, yet, it would be the most interesting is: which builds up more per unit of C assimilated (this would be an indication of which may contribute more to SOC building), or in other words a measure of enrichment/recovery.

7. Glucose is likely to behave differently to other C substrates, therefore I would restrict the title and Discussion to glucose (not carbon).

8. The rationale for the difference found in turnover time between the cytosol and PLFA is really not convincing and not well supported. Both active and dormant organisms have a membrane and if they're not active they wouldn't be picking up the 13C.

9.The explanation for why more 13C was in the bacterial than non-bacterial lipids completely ignores anything about their ecology or physiology.

10.Describing what fatty acids were more or less abundant (section 3.3.) is not really informative as these data don't reflect absolute abundances and these patterns of abundance are more or less the same for a lot of soils.

11. I would replace the term 'incorporation' with 'recovery' which seems to be what they calculated.

12.I find it surprising that soil moisture would be "essentially constant" across 50 days.

Minor methods comments The rational for the amount of C added is not presented. Not clear if the field collected soil column had or not vegetation. I don't understand the "assignment of fatty acids to distinct microbial groups by factor analysis".

---

## Author Comment (AC2) · 20 Aug 2016

**Thank you for the study evaluations.**

1. Hypothesis 1 could not really be tested given the duration and intervals of sampling during the experiment. Glucose gets processed, incorporated, lost and recycled into PLFA already in the first two days (e.g. Ziegler 2005) and this can vary with soil and environmental conditions. Starting measurements after 3 days leaves us without any information of when the peak of uptake took place (thus when time zero for decomposition started) and when recycling started which would matter for trying to estimating turnover. Then on the other end, 50 day was not sufficient time for the aminosugars to finish building up and start decomposing, as the authors discuss. Perhaps the data can be used to answer a different question. From Figure 2, we don't know how good

the model fits were (it would be important given there were three points and large error bars).

**Thank you for the comment. The time 0 was not taken into account, due to several reasons: i) microorganisms uptake glucose from soil solution very fast according to lab observations, (reported times are in the second to minutes range (Hill et al., 2008), however it can be slower for the field conditions. So, due to part of the glucose still can be in soil solution the 13C incorporation into cytosol will still increase, whereas on day 3 the complete uptake of glucose can definitely occur. ii) Due to glucose is taken up first into the cytosol pool and incorporation into other pools is delayed, that it would not be possible to have the t0 for other pools. Moreover, in our experiment we were focus on the 2nd phase of glucose utilization and not on the first (uptake phase). Figure 2: we agree that this is not big amounts of points, but curve fitting was done for 4 replications for each time point. Due to this parameter is related to the big uncertainty not a lot of conclusions are made base on that parameter. The most important is that this is only approach, and also that the 13C incorporation decrease or increase over time for the pools which can be clearly seen based on the 3 points. Moreover our experiment was done in the field and much larger variations can be expected compare to lab experiment were the soil is homogenized before the experiment (before the application of tracers).**

2. Hypothesis 2 is about differential incorporation by bacteria and fungi (who incorporates it faster). Again, missing the first three day is pretty critical (Ziegler 2005 clearly shows this). There are already many experiments that have assessed "initial" incorportation of 13C into biomarker lipids and we wouldn't then need 50 days incubation for this. Also, I am surprised that there was no effect of time on the composition of the PLFA as 50 days is quite a long time for microbes and PLFA profiles tend to be more sensitive to time than any other driver, and, C depletion in 50 days of an incubation would be substantial.

**Thank you for the comment. Yes, we agree that the time is very important in the labeling experiments, especially if the labile substances are applied. However, our experiment was done in the field (with 1.5 kg of soil and changes of the condition between day and night), and microorganisms can take up glucose much slower compare to lab experiments. In this case measurement of 13C in the PLFAs on the 1st or 2nd day will probably not show the highest incorporation. That is why we did not make the 1st point 1 day after 13C incorporation. Concerning the initial incorporation: most of the experiments were done in the lab, whereas our experiment was done in the field, were processes goes much slower. The data on total remaining 13C after 50 days of experiment are clearly show this: around 70% of 13C derived glucose remained, showing slow rates of processes in the field experiment. Moreover, we were focused on the 2nd phase of glucose utilization and not on the first. The structure of PLFAs stayed quite stable during the field experiment because of the following: i) soil was plowed two week before the experiment and microorganisms were already adopted to the changed conditions of limiting C and ii) low concentration of glucose were applied, which did not promote microbial growth or changes of microbial community structure.**

3. Hypothesis 3 is hard to follow. I tried but was not able to understand it.

**Thank you for the comment. We have deleted hypothesis 3.**

4. In the Introduction, the paragraph comprising Lines 96-108 is a very convoluted and hard to follow, however, it refers to the main rational for carrying out this study.

**This paragraphs will be improved.**

5. I don't understand why there was not an attempt to estimate ks for the PLFAs. That would be the main purpose of this approach, in my view. Also, how can the VAM be building up 13C, if they are mycorrizhae? This probably suggest the marker was indicating a saprotroph, not mycorrhizae, which is known to happen in soils.

**The decomposition constants can be estimated in our study, however, only for the groups where 13C enrichment decreased. However, we aware that these values can not be obtained for all groups and thus, we can not make quality comparison. Yes, we**

agree that 16:1w5 can belong to saprotrophic fungi. This will be corrected in the text.

6. About the aminosugars, what we still don't know and doesn't get explored and discussed, yet, it would be the most interesting is: which builds up more per unit of C assimilated (this would be an indication of which may contribute more to SOC building), or in other words a measure of enrichment/recovery.

**Thank you for the comment. We can not estimate how much aminosugars were build per unit of C assimilated, because it is not possible to separate amino sugars of living microorganisms from one accumulated in SOM in composition of microbial residues. Only what we did, we estimated enrichment of amino sugars by 13C on the day 3: " To estimate the 13C enrichment into amino sugars of living cells, we first calculated the amount of amino sugars in the living MB pool based on the fatty acids content. Assuming that PLFAs are present only in living biomass, and that the ratio of fatty acids to amino sugars in living biomass is about 0.23 (Lengeler et al., 1999), we estimated the amount of amino sugars in living MB to be 0.20 $\mu$mol g-1 soil fatty acids/0.23 = 0.87 $\mu$mol g-1 soil. The estimated percentage of amino sugars in living biomass from the total amino sugar pool was 0.87/7.70 (total AS ($\mu$mol g-1 soil))*100 = 11%. This estimate agrees with that of Amelung et al. (2001a) and Glaser et al. (2004), who reported that the amount of amino sugars in living biomass is one to two orders of magnitude lower than in the total amino-sugar pool. We calculated the 13C enrichment in amino sugars for the first sampling point, assuming that all replaced C is still contained within living MB after three days of glucose C utilisation. Total tracer incorporation into amino sugars consisted of 0.00071 $\mu$mol glucose derived 13C in amino sugars g-1 soil/0.87 ($\mu$mol amino sugars g-1 soil)*7 (mean amount of C atoms in amino sugars)*100 = 0.57% of the C pool. Comparison of these data with the 13C enrichment into PLFAs and the cytosol allowed us to conclude that the replacement of the amino sugar C with glucose derived 13C in living biomass is two-fold slower than the replacement in PLFAs, and faster than in the cytosolic pool". This is presented in the discussion section.**

7. Glucose is likely to behave differently to other C substrates, therefore I would restrict the title and Discussion to glucose (not carbon).

**Title was corrected: Glucose C turnover in cell compartments and microbial groups in soil. The discussion will also be corrected and only glucose C will be discussed.**

8. The rationale for the difference found in turnover time between the cytosol and PLFA is really not convincing and not well supported. Both active and dormant organisms have a membrane and if they're not active they wouldn't be picking up the 13C.

**Thank you for the comment. The idea of the work was estimate C turnover time in various cell microbial pools: cytosol, PLFAs and amino sugars. Cytosol is always assume as the most labile pool and supposed that C here should have the fastest turnover. However, in our work it is clearly seen that is it not the case, and actually C turnover times here are much slower that in structural cell pools (such as PLFAs). Our data on 13C pool enrichment include information as about dormand as about active microorganisms (due to calculation was done relatively to the total C pool), and clearly show that more 13C is incorporated into cell membranes than in cytosol and also that C turnover is slower in cytosol, than in PLFAs. Concerning dormancy: even a dormant microorganisms repair their membranes and, thus, have a C turnover in them, whereas microorganisms can be survived without DNA reparation.**

9.The explanation for why more 13C was in the bacterial than non-bacterial lipids completely ignores anything about their ecology or physiology.

**Thank you for the comment. The goal of the study was not to investigate the ecology or physiology of microorganisms, and for that purpose only low concentrations of glucose were applied. We were mainly concentrated on the turnover times of C in various cell pools and worked with microbial community which was already formed in the soil before we started the experiment.**

10.Describing what fatty acids were more or less abundant (section 3.3.) is not really informative as these data don't reflect absolute abundances and these patterns of abundance are more or less the same for a lot of soils.

**Thank you for the comment. We will reduce this section.**

11. I would replace the term 'incorporation' with 'recovery' which seems to be what they calculated.

**Thank you for the comment. This will be changed in all entire text.**

12. I find it surprising that soil moisture would be "essentially constant" across 50 days.

**The data for soil moisture are the following: $25.7\pm1.2$ (3 days), $23.3\pm1.3$ (10 days), $21.4\pm0.7$ (50 days). This information will be added into the material and method section. Due to we did not sample the soil directly after the rain, the moisture variations were low.**

Minor methods comments The rational for the amount of C added is not presented.

**The C was added in the amount to prevent any priming effects, as well as growth of microorganisms due to glucose application. The concentration was chosen to trace the natural pool of glucose in soil solution (Fischer et al., 2007), rather than stimulate the activity of microorganisms.**

Not clear if the field collected soil column had or not vegetation.

**The columns had no vegetation by the collecting time, as well as when the 13C glucose was applied. This will be added into materials and methods section.**

I don't understand the "assignment of fatty acids to distinct microbial groups by factor analysis".

**Factor analysis was used as classification method, with the main purpose to split the microorganisms which belong to one group (according to literature) into subgroups which behave differently in the soil. Based on the analysis, fatty acids which were**

loaded into one factor with the same sign (+ or -) and were related to the one bacterial or fungal group, were summed up together. In the end of the analysis only groups of PLFAs related to ether bacteria or fungi were presented. The information about which fatty acids belong to which microbial groups was taken from the literature (Zelles 1997).

---

## Author Comment (AC3) · 20 Aug 2016

The authors of this manuscript have analyzed the turnover of different cellular compounds/fractions for different microbial groups using a 13C labeling experiment (3, 10 and 50 days). This is clearly a worthy and important goal. The experiment is done well although the number of harvests (3) is minimal for this determination of turnover. For reasons described below, I think the manuscript is not acceptable for publication in its current form. The goal of the manuscript is to evaluate the turnover time of C in each pool and to assess the contribution of bacteria and fungi to SOM. A second goal is to determine the turnover time for different categories of microbes. They hypothesize that turnover time is short for cytosol, intermediate for PLFAs, and long for amino-sugars. However, the results they find indicate that turnover time of lipids<aminosugars<cytosol. They hypothesize that, based on aminosugar ratios, the

bacteria contribute more to SOM than fungi, however, the results are contradictory (one ratio suggests bacteria, the other fungi). Instead of defending these observations and rejecting the hypotheses, complex reasons are proposed why turnover time of the cytosol is long, but it is still a "labile pool" that turns over fast but has tight cycling, and, in the discussion, it turns out that one of the aminosugar ratios is "better" than the other, so that the bacterial contribution to SOM is high. In other words, experimental results could not cause rejection of the hypotheses, therefore I have to conclude the experiment was poorly designed and not able to test the proposed hypotheses.

**Thank you for evaluation of our study. We have defended our observations, concerning cytosol, because numerous previous data have reported much shorter turnover times of C. Moreover, cytosol is assumed as a labile pool, and we needed to explain why our initial hypothesis was rejected. Concerning amino sugars: application of ether galactosamine/muramic acids ratio or glucosamine/ muramic acids is a still a topic of discussion in the soil science community, however, according to reviewer recommendations we will exclude this from the discussion and will shorten this section significantly.**

There are several reasons for the inability of this experiment to deliver results that are strong enough to test the hypotheses 1) It is unclear what "cytosol" is and why it is thought to be labile (L37). Although aminosugars and PLFAs are (bio)chemically distinct, this is less so for the fraction "cytosol" (L121, L179 and following). In order to understand the differences between lipids, aminosugars and "cytosol", the authors will have to analyze the amount of lipids and aminosugars in the cytosol fraction.

**Thank you for the comment. Term "cytosol" was used by us for shortness. This is pool extracted by 0,05M K2SO4 after the chloroform fumigation and substraction amount of C extracted from the soil by 0,05M K2SO4 without fumigation. The reasons why cytosol is assumed as a labile pool due to : i) it contains significant amounts of carbohydrates (20-30%) (Joergensen et al., 1996), ii) it is a pool where the main chemical processes of the cell occur. Moreover, numerous researches have obtained much shorter turnover times of C in, than was obtained in our study. It is not possible to measure lipids in the**

cytosol pool, because lipids are not dissolved in 0,05M K2SO4 which is used to extract cytosol pool. Pool of amino sugars in cytosol fraction is several orders of magnitude lower than one extracted by acid hydrolysis.

2) The experiment was not long enough to calculate turnover time for aminosugars (Fig. 2; ). Moreover, although turnover is calculated using one exponential declining function (Fig. 2), in the discussion, a whole paragraph is dedicated stating that glucose decomposition is bi-phasic (L 362), and so the use of a single exponential function needs to be defended.

**Thank you for the comment. With the time scale which we have investigated, we were mainly focused on the second phase of glucose utilization. So, that is why the calculated half -life times of C are related to the second phase, and that is why we have used the single exponential model. Moreover, 3 time points do not enough for the double exponential model use.**

Furthermore, conclusions about turnover rates are presented for PFLAs and aminosugars, without numbers to back up the conclusions. This is because of increasing 13C contents with time for aminosugars and fungal PLFAs; however, if the turnover times cannot be calculated, the conclusion should not be drawn, data should not have been presented (under this title) and/ or more data should be collected.

**Thank you for the comment. If the 13C replacement in amino sugars and fungal PLFA increase, it is obviously that the C turnover is longer, compare to the pools, where 13C replacement started already decrease within the 50 days of experiment. However, we clearly wrote this problem in the discussion section and did not to any specific conclusion about the data which we did not calculate. The title of the ms can be changed to: "Glucose C turnover in microbial cell compartments in soil", to avoid confusion for the readers.**

Additionally, turnover rates should have been calculated for the various bacterial and fungal groups based on PFLA data (according to the title). Finally, the presented

turnover rates are presented without an estimate of the error associated with it (for example R2 value in Fig. 2, 3 and 4, SE for the turnover time values), making it impossible to evaluate whether the estimated turnover times for lipids and cytosol are truly different.

**Thank you for the comment. Due to some PLFAs showed still increase in 13C replacement, we can calculate the C turnover rates only for the some PLFAs (which showed decrease in 13C replacement). However, in that case we can can not compare C turnover times for all groups and make a correct conclusions. However, if reviewer insist, we will make calculations for the groups, which showed decrease. The R2 values will be provided for the figure 2.**

3) Hypothesis 1 is interesting, but cannot be tested in this experiment, as the initial uptake and incorporation in cytosol and other pools is fast. For example, Frey et al 2013 show that glucose uptake and incorporation in microbial "cytosol" occurs within 6 hours. The authors need to explain why and how this hypothesis can be tested using the experiment they designed. Hypothesis 3 is not a hypothesis but a (simplifying) assumption, used to interpret the results of this study, not a testable hypothesis. Moreover, the assumption is by definition wrong, but at best is an acceptable approximation. However, no evidence is given to support this assumption. Is 50 days incubation still short enough that no aminosugars are transferred to the necromass pool? In general, the hypotheses are poorly defended or explained mechanistically.

**Thank you for the comment. We agree with reviewer and removed hypothesis 3. We have improved hypothesis 1 and focus only on the turnover times: we hypotheses that 1) turnover times of pools follow the order cytosol<PLFA<amino sugars, because substances taken up by cells first are transported by membrane proteins into cytosol, from where they get distributed to other cellular pools. Moreover we have investigated the second phase of glucose C utilization, and not initial uptake, which was studied already many times.**

Additional general comments The statistics need to be further developed. The estimates of the turnover for the different fractions/compounds (L 304) need to be described with a mean and error. R2, significance and SE need to be added with Fig 2, 3 and 4. Current description does not make it possible to verify the assertion of the authors that the turnover rates of the various pools are significantly different. Fig. 5 does not add to understanding or interpretation of the results and can be removed.

**Thank you for the comment. Significance and SE are in the figures 2, 3 and 4. R2 will be added to the figure 2. Figure 5 is a synthesis of the ms results, however, if reviewer insist we will remove it.**

The observation that the 13C incorporation (as a percentage) was higher in PLFA than in cytosol does not logically result in a conclusion that the incorporation is faster (L32). This result may just be a reflection of the size of the pool (PFLA versus "cytosol"), and certainly does not show "the importance" of membranes "for initial C utilization".

**Thank you for the comment. On the L32 we speak about enrichment of pool by 13C - means % of 13C in the total pool C, and this completely account for the pool size. The incorporation of 13C does not account for the pool size, but we do not speak in L32 about that.**

The use of the term filamentous organisms should be avoided. The authors probably mean fungi. I like the intent of L46, however, the comparison of the dynamic behavior of the three pools remains poorly developed.

**Thank you for the comment. The term filamentous organisms can not be avoided, because we speak about fungi and actinomycetes (which can not be named as fungi). The dynamic behavior of the three pools will be improved.**

Careless use of references: L 68: Malik et al have not reported on cytosol, nor on its supposed heterogeneity. It is not at all clear how location would affect the turnover time of membranes and cell walls (L70).

**Thank you for the comment. Malik et al investigated pool extracted after chloroform fumigation, which is partly reflect the composition of a cytosol pool. He measured molecular mass distribution and found out many fraction with different molecular weights, which is an observation of fraction heterogeneity. The tern location will be corrected to the term "function".**

Bremer and Kuikman (1993; L 73) are not experts in microbial physiology, and therefore not an authorative source to support the statement that labeled glucose appears first in the "cytosol". In fact, they only looked at the cytosol (fumigation-extractable) so cannot comment on whether other compounds or fractions become labeled first or later.

**Thank you for the comment. We will provide other references: Bacterial Metabolism (G. Gottschalk, 1986).**

A reference is needed to support the assumption that "the cytosol is considered to be the most dynamic pool within microbial cells". Furthermore, heterogeneity (L75-76) has never stopped any calculation of turnover times, as is evident in soil organic matter turnover studies. Important references are missing for example those by Malik et al 2015 where comparison between "cytosol" and PFLAs are made (and DNA/RNA).

**Thank you for the comment. The reference Malik et al 2015 will be added. L75-76 we mean, that due to many different compounds are in cytosol composition (with different molecular masses) the turnover time of C in this pool is a mean of turnover times of these components. We have corrected: Organic compounds that are taken up by microorganisms first enter the cytosol (Bremer and Kuikman, 1994), which has a high heterogeneity in composition.**

L96 and following: This paragraph tries to distinguish between cellular turnover – I assume as a consequence of cell death is what is referred to here – and turnover of compounds within a living cell. However, it is not that easy to make that distinction – how does one distinguish between lipids being recycled and reused, taken apart and made into for example amino-acids, while other amino acids are recycled into lipids,

and what happens after cell death – uptake of lipids by other organisms intact incorporated, reused, recycled, taken apart and/or turned into CO2. Moreover, the observations of increasing 13C concentrations for fungi versus decreasing ones in bacteria suggests some transfer of compounds, but remains unexplained in this manuscript.

**Thank you for the comment. We will deep this problem in introduction section, and will provide some possible explanations of 13C increase in fungal PLFA (cross feeding, recycling of 13C from microbial metabolites).**

L 146: it is not clear to me why unlabeled glucose was added to the control treatments.

**Thank you for the comment. Unlabelled glucose was added to make the experimental conditions equal (C content of the additions). If we add only water, than conditions of the experiment would not be the same and we would not have true controls.**

L 149: explain why the shelter was put in place and why it was removed. What was the effect of this on the soil moisture content?

**Thank you for the comment. Shelter were putted to avoid the rain and the flow of glucose from the soil columns. The soil moisture remained nearly constant. The shelter were removed because this was field experiment and natural conditions should present on it.**

L 157: why was the soil stored at 5 °C for 5 days prior to chloroform-fumigation analysis? What happened to the "cytosol" during that time? Does this mean that the value for cytosol is really the value after 8, 15, and 55 days?

**Thank you for the comment. This is standard storage temperature for the soil sample before the analysis, if they can not be performed in the same day as sampling day. No, due to temperature in the field was around 15°C, the storage at 5°C will not cause strong effect on the cytosol fraction.**

L 180: defumigation is not a word.

**Thank you for the comment. Was changed to: "After removing the rest of chloroform from the soil....".**

L 186: "extraction efficiency" not "extraction factor"

**Thank you for the comment. Was changed. L 247: "the assignment of fatty acids to microbial groups ..." is confusing me. Does this mean that as part of this study, biomarker PFLAs are assigned to group independent of what is done in other studies?**

**Thank you for the question. No, it means that we used previous studies to assign PLFAs measured in our experiment to the particular microbial groups. The appropriate reference is provided (Zelles, 1997).**

L 249: this procedure is not clear to me, but I am not at all familiar with PFLA/Microbial community analysis. My first impression was that the analysis is basically a community analysis – showing, based on PFLAs, what the community looks like. However, L 247 suggests that with this procedure, PFLA are assigned to microbial taxa, but then in the heading of Supplementary Table 1 it suggests that literature data is used – Please clarify what the table is used for, how (and what) literature data is used, and what the results of this analysis means for your experiment. Similarly, L 431: the arguments for using the 16:1w5 as a biomarker for VAM and not G- are weak. The abundance of VAM needs to be expressed relative to G- bacteria. Table one suggest that the total C for PFLAs is higher than for VAM, thus is more abundant (?).

**Thank you for the comment. The procedure is the following: PLFA analysis provides content of various PLFAs, some of them are specific for the groups of G+,G- bacteria, fungi, actinomycetes and arbuscular mycorrhizal fungi. These specifity was established based on the analysis of pure cultures (Zelles, 1997). Based on the factor analysis table of factor loadings was obtained, and within one factor the fatty acids with the same sign (+ or -) and which are belong to one group (base of the table provided in Zelles, 1997) were related to one specific group and content of their PLFAs were summed up. With PLFA analysis is not possible to conclude about the abundance of VAM and G-**

bacteria. Only the approximate calculated coefficient was proposed in literature (Baath and Anderson, 2003) for fungal biomass. That is why we spoke only about the content of biomarkers in our study, and not about biomass of microorganisms. Based on the PLFA-C it is not possible to conclude about the biomass of microbial groups.

L290: the description of the results (declines between 3 and 10 days but then remains constant then constant) does not match the assumed exponential decline. Please explain.

**Thank you for the comment. The information provided at the L290 is about recovery of the tracer in various pools, whereas exponential decline is seen for the enrichment of C in the pools (portion of 13C in the total C pool).**

Fig. 2, 3, 4: the statistical tests should also be done between harvests, not only between microbial groups.

**Thank you for the comment. The statistical tests will be added, where the differences between sampling points are clearly seen.**

L 347: the explanation for the differences between this study and published results, namely the amount of glucose added and the microbial activity, are not revealed. Some further information on these explanatory variables would be appreciated. Is microbial activity measured in this study, microbial activity is not measured? The idea that microbes store glucose when added in small quantities is unproven – it is a mere assumption, recently defended by Sinsabaugh et al 2013, but evidence for storage was absent in recent experiments by Dijkstra et al (2015). The idea that the storage leads to maintenance is in contradiction to the 80% recovery after 50 days, and with the idea that microbial pools and cells turn over fast.

**Thank you for the comment. More information about the effect of amount of added glucose will be added. No, microbial activity was not measured. We agree with the reviewer and remove maintenance from the ms. We mean rather incorporation into cell**

components than mineralization.

L 362: the description of the two stages of glucose decomposition – 1) CO2 production plus biosynthesis, and 2) C incorporated in microbial cells is used for anabolism is confusing. Is anabolism different from biosynthesis? Is during the second phase CO2 production absent? How do the two phases relate to the biosynthesis of lipids, cytosol, and aminosugars? Please clarify

**Thank you for the comment. We will use only the term biosynthesis. We wanted to highlight that during the first phase of glucose utilization the C is mainly goes to CO2, whereas during the second phase more C goes for biosynthesis, which includes reuse of cell compounds. In our study we were focus on the second phase of glucose utilization.**

L 395: what is this model, please explain some salient details and how it agrees with your observations.

**Thank you for the comment. In revised version we will remove these sentences.**

L 419: this rationalization needs some references or evidence that contact with the environment leads to rapid turnover.

**Thank you for the comment. Typical example for that is fast response of cell membranes to stress conditions (water stress, pH stress, pollution, lack of available C): formation of cyclo-fatty acids in membranes of gram-negative bacteria (Bossio et al., 1998, Guckert et al., 1986, Kieft et al.,1997).**

L 421: the problem of active and inactive cells for cytosol dynamics is similar for lipid dynamics, as inactive cells also have membranes.

**Thank you for the question. Actually the lipid behavour is different from the cytosol pool: even a inactive cells repair membranes, whereas other cell pool not necessary to be repaired. The typical example for that is: dormant microorganisms live with damaged DNA, but never with damaged membranes.**

L 482: how is this conclusion drawn when the turnover rate cannot be calculated according to L486. L 506: how do you determine that the turnover of the amino-sugars is higher than that of the cytosol pool? L 509: this would be a wonderful conclusion, but it does not appear in the abstract at all. What is the reason that the cytosol is so stable? Please elaborate.

**Thank you for the comment. Due to C replacement in amino sugars pool is still increasing after 50 days, we can not calculate the C turnover times. However, if replacement is still increasing it is obviously that C turnover in amino sugars is slower than in PLFA and cytosol. The conclusion about: " This reflects that microbial C turnover is a phenomenon that is not restricted to the death or growth of new cells, but that even within living cells, highly polymeric cell compounds, including cell walls, are constantly replaced and renewed" will be included into abstract. The reason that cytosol is stable is that in contains compounds with different molecular size (Malik et al., 2013): as low molecular weight, which renew fast and high molecular weight which renew slow.**

L 511 and following – the results from the measurements seem to indicate contrasting conclusions – bacteria or fungi are most important (L516 and following). It is then stated that only the galactosamine/muramic acid ratio should be used. So, this means that the reader has wasted a number of valuable brain cells thinking about the galactosamine/glucosamine ratios, and looked at the data, but that was all a waste of time? Why not start with what is known (galac/muramic ratio) and leave it at that. Furthermore, there is a lot more text about the three aminosugars and their ratios in relation to bacteria and fungi – is that still relevant inlight of L 521?

**Thank you for the comment. We agree with reviewer, this par of discussion will be reduced, and only galac/muramic ratio will be presented.**

Fig 1: explain what is total 13C remaining, what is non-specified pool? Remake the Fig so that the SE of the aminosugars are fully shown.

**Thank you for the comment. Total 13Đą remaining is a rest amount of 13C measured**

[Figure]

in bulk soil, it is a sum of 13C in cytosol, PLFAs, amino sugars and non-specified pool. Non-specified pool is amount of 13C rest in the soil without 13C incorporated into cytosol pool, PLFAs and amino sugars. This explanation will be added into the material and method section.

Fig. 2: what is the equation with the word "replacement" in it? I think it is just the function of 13C over time, and thus the word replacement can be removed, but I may be wrong. Add R2, P value and significance (and SE of the turnover estimate)

**Thank you for the comment. This is enrichment, this is misprinting. This will be corrected in the paper.**

Fig. 3: instead of showing differences between microbial groups, we need to know the differences between dates AND microbial groups to evaluate how these differences represent significant differences in turnover, and whether this turnover differs between groups. Moreover, the goal of this paper was to determine differences in turnover between microbial groups, but this is not calculated. If turnover cannot be calculated for groups where 13C enrichment is increasing over time, what was the basis for the conclusion that turnover differed between fungi and bacteria (L320)?

**Thank you for the comment. The differences between dates will be provided. Turnover of the different groups can be calculated only for one, which have decrease in 13C replacement. Conclusion about the differences in turnover time of C between fungi and bacteria is made based on the trend of 13C replacement: if 13C replacement is still increasing it means, that the C turnover in particular PLFAs is longer compare to one where 13C replacement decreasing within the experimental time.**

Fig 5: not really helpful.

**Thank you for the comment. Fig. 5 was strongly improved: fluxes were clearly marked by the different size of arrows, position of x and y axises were changed. However, if reviewer insist we will remove it.**

---

## Author Response (AR1)

Revision of the Ms. No. Biogeosciences Discuss., doi:10.5194/bg-2016-214, 2016 Glucose C turnover in cell compartments and microbial groups in soil

Dear Prof. Pendall,

we are very thankful to the reviewers for their helpful suggestions and recommendations. We carefully improved the ms according to their comments and incorporated nearly all their suggestions.

Please find below the response to the reviewers' comments (in black) and the improved ms.

We hope that the ms fulfill the requirements of Biogeosciences.

With kind regards, Anna Gunina and co-authors.

**Reviewer 1.**

I have revised ms CARBON TURNOVER IN CELL COMPARTMENTS AND MICROBIAL GROUPS IN SOIL by Gunina et al. To reveal the contribution of particular microbial groups to C utilization and C turnover within the microbial cells, the fate of 13Clabeled glucose was studied under field conditions in this work. The 13C was traced in cytosolic substances, amino sugars and phospholipid fatty acids (PLFA) at intervals of 3, 10 and 50 days after glucose addition. The strong and positive site of this manuscript is the combination of 13C labeling with the subsequent analysis of several microbial cell compartments and biomarkers as a unique approach to understanding C partitioning within microbial cells and the microbial communities in soil. This knowledge is crucial for both assessing C fluxes and their recycling in soil, and studying the contribution of C from microbial residues to SOM. There are some weaknesses of the manuscript, but they do not affect too much on the paper quality. Below I have included general and specific comments regarding all sections, and after all I recommend to authors revise, clarify and substitute some information through the whole text. To conclude, I think present manuscript may be published in Biogeoscience journal.

**Thank you very much for the evaluation of the ms!**

**General comments**

Materials and methods: There is no information on bulk density of the soil in the paper. Provide this information, as it is not clear how you calculated soil mass in cylinder

**The bulk density is 1.36 g cm-3. This was be added into materials and method section.**

Check all links to figures, equations and tables and present them whether in full text or shortened form (fig, eq., etc), but the same through the whole manuscript.

**Thank you for the comments. Corrected.**

Specific comments

1

L 180 – 183 I recommend to change "due to "in "Due to fumigation–extraction technique allows. . ..." for "As fumigation–extraction technique allows. . ..."

**Was corrected: " As fumigation–extraction technique allows to obtain not only soluble components, but also cell organelles and cell particles, we named pool of C in fumigated extracts as cytosol only for simplification of terminology."**

L 186 Revise sentence: In After organic C concentration were measured  $\rightarrow$  After organic C concentrations were measured OR After organic C concentration was measured

**Corrected " After organic C concentrations were measured, the  $K_2SO_4$  extracts were freezedried and the  $\delta^{13}C$  values....".**

L 295 Clarify if it was 50th day of experiment or after 50 days in the sentence "13C in amino sugars increased two fold during the 50 day experiment (p<0.05)"

**Corrected: " In contrast,  ${}^{13}$ C in amino sugars increased two fold on the 50th day of experiment (p<0.05)".**

L 301-302 Paraphrase the sentence, as it is difficult to read: "The enrichment was the highest in PLFAs (Fig. 2) and was 5–8 times lower in the cytosolic pool. The 13C enrichment in amino sugars was the lowest (Fig. 2) "

**Corrected: "The pool enrichment was the highest for PLFAs and the lowest for amino sugars (Fig. 2)".**

L 427- 429 Substitute "we used agricultural soil" on "we conduct experiment on agricultural soil". In your paper you wrote about field experiment, not incubation one with agricultural soil. It is not correct "to use" soil in this situation.

**Thank you for the comment, it was corrected: "Because we conduct experiment on agricultural soil with pH close to neutral (6.6), the predominance of bacterial PLFAs was expected".**

L 450-451 Revise sentence "However, replacement of PLFAs C by glucose derived 13C is only a proxy of microbial activity and only partly confirms the real mechanisms."

**Corrected: "However, replacement of PLFAs C by glucose derived 13C is only a proxy of microbial activity and can only partly estimate the real activity of microbial groups".**

L 486-487 What long-term sampling points mean? Maybe, you suggest conducting long-term experiment for turnover time calculation? Or add sampling points to existing experiment?

**Corrected: "To calculate C turnover time in this pool, conducting of long-term experiments is necessary".**

L 526- 527 Substitute "our the site" on "experimental site or site" in "the environmental conditions of our the site, namely, long-term agricultural use. . .."

Удалено: ¶

**Corrected: "...the environmental conditions of the site, namely, long-term agricultural use, which promotes the dominance of bacterial communities".**

**Tables**

L 809 It will be better to write "Data present mean of three replicates" instead of "Data present mean of three time points".

**Corrected: "Data present mean of three time points (with four replications for each time point)  $\pm$  SE".**

**Figures**

L 887 It is difficult to read graphs for Ac. Expand the space between Ac and second scale for VAM and Fungi.

**Was corrected.**

**Reviewer 2.**

Anonymous Referee #2 Received and published: 26 July 2016 Contributing to the understanding the C turnover of different cell compartments is certainly a valuable goal. This manuscript reports the findings of a 3-50 day incubation experiment where 13C from glucose is followed into the cytosol, PLFAs and amino sugars. They set out to test three hypotheses: (1) turnover times increase in the order cytosol>PLFA>amino sugars; (2) incorporation of 13C is faster for bacterial than fungal markers; (3) "due to amino sugars have long turnover times and are mainly dominated in microbial necromass, all incorportated 13C can be related only to living biomass and allow estimate percent of replaced C in amino sugars of living organisms. I am concerned about the limitations brought by sampling intensity and analysis of the data and mostly that the findings that were possible with this study don't really allow for testing the hypotheses presented. Probably as a result. A lot of what's presented in the Discussion is not critical or essential and it's either general, repetitive or tangential.

**Thank you for the study evaluations.**

1. Hypothesis 1 could not really be tested given the duration and intervals of sampling during the experiment. Glucose gets processed, incorporated, lost and recycled into PLFA already in the first two days (e.g. Ziegler 2005) and this can vary with soil and environmental conditions. Starting measurements after 3 days leaves us without any information of when the peak of uptake took place (thus when time zero for decomposition started) and when recycling started which would matter for trying to estimating turnover. Then on the other end, 50 day was not sufficient time for the aminosugars to finish building up and start decomposing, as the authors discuss. Perhaps the data can be used to answer a different question. From Figure 2, we don't know how good the model fits were (it would be important given there were three points and large error bars).

**Thank you for the comment. The first reason, that time 0 was not taken into account, because uptake of glucose by microorganisms from soil solution can be slower for the field**

Удалено: ¶

conditions than in lab observations (minute range (Hill et al., 2008). Thus, part of the added glucose still can be in soil solution 1 day after addition and incorporation of 13C into cytosol will still increase. Thus, it would not be possible to calculate C turnover time in cytosol pool. The second reason is: even if glucose is completely taken by microorganisms, the glucose-derived C incorporation into other pools is delayed (glucose C can be stored in cytosol, and only later used for biosynthesis), and thus, it would not be possible to have the t0 for all cellular pools. The third reason is that our experiment was focused on the 2nd phase of glucose utilization (using in biosynthesis) and not on the first (uptake phase).

Figure 2: We agree that we do not have big amount of points, but curve fitting was done for 4 replications for each time point. Due to calculation of turnover time is related to the big uncertainty, not a lot of conclusions were made based on that parameter. Even there are some uncertainties with calculating the C turnover times, the increase or decrease of 13C incorporation into the cellular pools with time can evidence about slow or fast turnover of particular pools. Moreover our experiment was done in the field and, even the increasing the amount of points would not help to reduce the errors, because much larger variations can be expected in the field conditions compare to lab experiments, where soil is homogenized before the experiment.

2. Hypothesis 2 is about differential incorporation by bacteria and fungi (who incorporates it faster). Again, missing the first three day is pretty critical (Ziegler 2005 clearly shows this). There are already many experiments that have assessed "initial" incorportation of 13C into biomarker lipids and we wouldn't then need 50 days incubation for this. Also, I am surprised that there was no effect of time on the composition of the PLFA as 50 days is quite a long time for microbes and PLFA profiles tend to be more sensitive to time than any other driver, and, C depletion in 50 days of an incubation would be substantial.

**Thank you for the comment. Yes, we agree that the time is very important in the labeling experiments, especially if the labile substances are applied. However, our experiment was done in the field (with 1.5 kg of soil and changes of the environmental condition between day and night), and microorganisms can take up glucose much slower compare to lab experiments. In this case measurement of 13C in the PLFAs on the 1st or 2nd day will probably not show the highest incorporation. That is why we did not make the 1st point 1 day after 13C incorporation. Concerning the initial incorporation: most of the experiments were done in the laboratory, whereas our experiment was done in the field conditions, where processes go much slower. The data on total remaining 13C after 50 days of experiment are clearly show this: around 70% of 13C derived glucose remained, showing slow rates of processes in the field experiment. Moreover, we were focused on the 2nd phase of glucose utilization and not on the first.**

The structure of PLFAs stayed quite stable during the field experiment because of the following: i) soil was plowed two week before the experiment and microorganisms were already adopted to the limiting C conditions and ii) low concentration of glucose was applied, which did not promote microbial growth or changes of microbial community structure.

3. Hypothesis 3 is hard to follow. I tried but was not able to understand it.

**Thank you for the comment. We have deleted hypothesis 3.**

Удалено: ¶

4. In the Introduction, the paragraph comprising Lines 96-108 is a very convoluted and hard to follow, however, it refers to the main rational for carrying out this study.

**This paragraphs was improved: "Bacteria and fungi have various chemical composition, which strongly contributes to their turnover rates in soil: for bacteria it consists 2.3-33 days, whereas for fungi it accounts for 130-150 days (Moore et al., 2005; Rousk and Baath, 2007; Waring et al., 2013). Despite turnover of microorganisms directly effect the C turnover rates in intercellular compounds (cell membrane and cell wall biomarkers), this relationship has rarely been investigated so far. However, the comparison of C turnover for cell membrane and cell wall components can be used to characterize the contribution of various microbial groups to medium-term C utilisation and to the stabilization of microbially derived C in SOM".**

5. I don't understand why there was not an attempt to estimate ks for the PLFAs. That would be the main purpose of this approach, in my view. Also, how can the VAM be building up 13C, if they are mycorrizhae? This probably suggest the marker was indicating a saprotroph, not mycorrhizae, which is known to happen in soils.

**The decomposition constants can be estimated in our study, however, only for the groups where 13C enrichment decreased. However, we fear that these values can not be obtained for all groups and thus, we can not make quality comparison. Yes, we agree that 16:1w5 can belong to saprotrophic fungi. This was corrected in the text.**

6. About the aminosugars, what we still don't know and doesn't get explored and discussed, yet, it would be the most interesting is: which builds up more per unit of C assimilated (this would be an indication of which may contribute more to SOC building), or in other words a measure of enrichment/recovery.

**Thank you for the comment. We can not estimate how much amino sugars were build per unit of C assimilated, because it is not possible to separate amino sugars of living microorganisms from one accumulated in SOM. However, we estimated enrichment of amino sugars by 13C on the day 3: " To estimate the 13C enrichment into amino sugars of living cells, we first calculated the amount of amino sugars in the living MB pool based on the fatty acids content. Assuming that PLFAs are present only in living biomass, and that the ratio of fatty acids to amino sugars in living biomass is about 0.23 (Lengeler et al., 1999), we estimated the amount of amino sugars in living MB to be 0.20  $\mu$ mol g-1 soil fatty acids/0.23 = 0.87  $\mu$ mol g-1 soil. The estimated percentage of amino sugars in living biomass from the total amino sugar pool was 0.87/7.70 (total AS (µmol g-1 soil))\*100 = 11%. This estimate agrees with that of Amelung et al. (2001a) and Glaser et al. (2004), who reported that the amount of amino sugars in living biomass is one to two orders of magnitude lower than in the total aminosugar pool. We calculated the 13C enrichment in amino sugars for the first sampling point, assuming that all replaced C is still contained within living MB after three days of glucose C utilisation. Total tracer incorporation into amino sugars consisted of 0.00071 µmol glucose derived 13C in amino sugars g-1 soil/0.87 (µmol amino sugars g-1 soil)\*7 (mean amount of C atoms in amino sugars)\*100 = 0.57% of the C pool. Comparison of these data with the 13C enrichment into PLFAs and the cytosol allowed us to conclude that the replacement of the amino sugar C with glucose derived 13C in living biomass is two-fold slower than the replacement in PLFAs, and faster than in the cytosolic pool". This is presented in the discussion section.**

Удалено: ¶

7. Glucose is likely to behave differently to other C substrates, therefore I would restrict the title and Discussion to glucose (not carbon).

**Title was corrected: Glucose C turnover in cell compartments and microbial groups in soil. The discussion was corrected and only glucose C was discussed.**

8. The rationale for the difference found in turnover time between the cytosol and PLFA is really not convincing and not well supported. Both active and dormant organisms have a membrane and if they're not active they wouldn't be picking up the 13C.

**Thank you for the comment. The idea of work was to estimate C turnover times in various cell microbial pools: cytosol, PLFAs and amino sugars. Cytosol is always assume as the most labile pool and supposed that C here should have the fastest turnover. However, according to results of our study, C turnover time in cytosol are much slower that in the structural cell pools (such as PLFAs).**

Our data on 13C pool enrichment includes information about both dormant and active microorganisms (due to enrichment was calculated relatively to the total cytosol C pool), and clearly shows, that more 13C was incorporated into cell membranes than in cytosol. It also shows that C turnover is slower in cytosol, than in PLFAs. Concerning dormancy: even a dormant microorganisms repair membranes and, thus, have a C turnover in them, whereas microorganisms can be survived without reparation of other cell pools (such as DNA).

9. The explanation for why more 13C was in the bacterial than non-bacterial lipids completely ignores anything about their ecology or physiology.

**Thank you for the comment. The goal of the study was to estimate the turnover times of C in various cell pools for native microbial community, which was already formed in the soil before we started the experiment. It was not a goal to investigate the ecology or physiology of microorganisms.**

10.Describing what fatty acids were more or less abundant (section 3.3.) is not really informative as these data don't reflect absolute abundances and these patterns of abundance are more or less the same for a lot of soils.

**Thank you for the comment. We have deleted this paragraph.**

11. I would replace the term 'incorporation' with 'recovery' which seems to be what they calculated.

**Thank you for the comment. The term 'incorporation' was change to 'recovery'.**

12.I find it surprising that soil moisture would be "essentially constant" across 50 days.

**The data for soil moisture were the following:  $25.7\pm1.2$  (3 days),  $23.3\pm1.3$  (10 days),  $21.4\pm0.7$  (50 days). This information was added into the material and method section. Due to we did not sample the soil directly after the rain, the moisture variations were low.**

Minor methods comments The rational for the amount of C added is not presented.

**The C was added in the amount to prevent any priming effects, as well as growth of microorganisms due to glucose application. The concentration was chosen to trace the natural**

Удалено: ¶

pool of glucose in soil solution (Fischer et al., 2007), rather than stimulate the activity of microorganisms.

Not clear if the field collected soil column had or not vegetation.

**The columns had no vegetation by the collecting time, as well as when the 13C glucose was applied. This information was added into materials and methods section.**

I don't understand the "assignment of fatty acids to distinct microbial groups by factor analysis".

**Factor analysis was used as classification method, with the main purpose to split the microorganisms which belong to one group (according to literature) into subgroups which behave differently in the soil. Based on the analysis, fatty acids which were loaded into one factor with the same sign (+ or -) and were related to one bacterial or fungal group (based on literature), were summed together. In the end of the analysis only groups of PLFAs related to bacteria or fungi were presented. The relation of fatty acid to particular microbial group was taken from the literature (Zelles 1997). The information about assignment of fatty acids to distinct microbial groups was added into material and method section.**

**Reviewer 3.**

Review of manuscript "Carbon turnover in cell compartments and microbial groups in soil" by Gunina et

al.

The authors of this manuscript have analyzed the turnover of different cellular compounds/fractions for different microbial groups using a 13C labeling experiment (3, 10 and 50 days). This is clearly a worthy and important goal. The experiment is done well although the number of harvests (3) is minimal for this determination of turnover. For reasons described below, I think the manuscript is not acceptable for publication in its current form. The goal of the manuscript is to evaluate the turnover time of C in each pool and to assess the contribution of bacteria and fungi to SOM. A second goal is to determine the turnover time for different

categories of microbes. They hypothesize that turnover time is short for cytosol, intermediate for PLFAs, and long for amino-sugars. However, the results they find indicate that turnover time of

lipids<aminosugars<cytosol. They hypothesize that, based on aminosugar ratios, the bacteria contribute more to SOM than fungi, however, the results are contradictory (one ratio suggests bacteria, the other fungi). Instead of defending these observations and rejecting the hypotheses, complex reasons are proposed why turnover time of the cytosol is long, but it is still a "labile pool" that turns over fast but has tight cycling, and, in the discussion, it turns out that one of the aminosugar ratios is "better" than the other, so that the bacterial contribution to SOM is high. In other words, experimental results could not cause rejection of the hypotheses, therefore I have to conclude the experiment was poorly designed and not able to test the proposed hypotheses.

**Thank you for evaluation of our study. We have defended our observations, concerning cytosol, because numerous previous data have reported much shorter turnover times of C in this pool. Moreover, cytosol is assumed as a labile pool, and we needed to explain why our**

Удалено: ¶

initial hypothesis was rejected. Concerning amino sugars: application of ether galactosamine/muramic acids ratio or glucosamine/ muramic acids is a still a topic of discussion in soil science community. However, according to reviewer recommendations we excluded this from the discussion.

There are several reasons for the inability of this experiment to deliver results that are strong enough to test the hypotheses 1) It is unclear what "cytosol" is and why it is thought to be labile (L37). Although aminosugars and PLFAs are (bio)chemically distinct, this is less so for the fraction "cytosol" (L121, L179 and following). In order to understand the differences between lipids, aminosugars and "cytosol", the authors will have to analyze the amount of lipids and aminosugars in the cytosol fraction.

**Thank you for the comment. Term "cytosol" was used by us for shortness. This pool is extracted by 0,05M K2SO4 after the chloroform fumigation. After measurement the C content in extracts from fumigated soil, we have substracted the amount of C extracted from the soil by 0,05M K2SO4 without fumigation. The reasons why cytosol is assumed as a labile pool due to : i) it contains significant amounts of carbohydrates (20-30%) (Joergensen et al., 1996), ii) it is a pool where the main chemical processes of the cell occur. Moreover, numerous researches have obtained much shorter turnover times of C in, than was obtained in our study. It is not possible to measure lipids in the cytosol pool, because lipids are not dissolved in 0,05M K2SO4 which is used to extract cytosol pool. Pool of amino sugars in cytosol fraction is several orders of magnitude lower than in acid hydrolysis fraction.**

2) The experiment was not long enough to calculate turnover time for aminosugars (Fig. 2; ). Moreover, although turnover is calculated using one exponential declining function (Fig. 2), in

the discussion, a whole paragraph is dedicated stating that glucose decomposition is bi-phasic (L  $\,$

362), and so the use of a single exponential function needs to be defended.

**Thank you for the comment. With the time scale which we have investigated, we were mainly focused on the second phase of glucose utilization. So, that is why the calculated half -life times of C are related to the second phase, and that is why we have used the single exponential model. Moreover, 3 time points do not enough for the double exponential model use. We have shorten the discussion about bi-phasic glucose decomposition.**

Furthermore, conclusions about turnover rates are presented for PFLAs and aminosugars, without numbers to back up the conclusions. This is because of increasing 13C contents with time for aminosugars and fungal PLFAs; however, if the turnover times cannot be calculated, the conclusion should not be drawn, data should not have been presented (under this title) and/ or more data should be collected.

**Thank you for the comment. Taken into account that,the 13C replacement in amino sugars and fungal PLFA increased, it can partly prove that the C turnover times in these pools are longer, compare to the pools, where 13C replacement started to decrease within the 50 days of experiment. We clearly wrote the problem about calculating the C turnover time for amino sugars in the discussion section, and did not do any specific conclusions about the data which we did not calculate. Additionally, turnover rates should have been calculated for the various bacterial and fungal groups based on PFLA data (according to the title). Finally, the presented turnover rates are presented without an estimate of the error associated with it (for example R2 value in**

Fig. 2, 3 and 4, SE for the turnover time values), making it impossible to evaluate whether the estimated turnover times for lipids and cytosol are truly different.

**Thank you for the comment. Due to some PLFAs showed increase in 13C replacement with time, we can calculate the C turnover rates only for the some PLFAs (which showed decrease in 13C replacement). However, in that case we can not compare C turnover times for all microbial groups and can not make correct conclusions. However, if reviewer insists, we will make calculations for the groups, which showed decrease.**

3) Hypothesis 1 is interesting, but cannot be tested in this experiment, as the initial uptake and

incorporation in cytosol and other pools is fast. For example, Frey et al 2013 show that glucose

uptake and incorporation in microbial "cytosol" occurs within 6 hours. The authors need to explain why and how this hypothesis can be tested using the experiment they designed. Hypothesis 3 is not a hypothesis but a (simplifying) assumption, used to interpret the results of

this study, not a testable hypothesis. Moreover, the assumption is by definition wrong, but at best is an acceptable approximation. However, no evidence is given to support this assumption.

Is 50 days incubation still short enough that no aminosugars are transferred to the necromass pool? In general, the hypotheses are poorly defended or explained mechanistically.

**Thank you for the comment. We agree with reviewer and removed hypothesis 3. We have improved hypothesis 1 and focus only on the turnover times: we hypothesised that 1) turnover times of C in pools follow the order cytosol<PLFA<amino sugars, because substances taken up by cells first are transported by membrane proteins into cytosol, from where they get distributed to other cellular pools. Moreover we have investigated the second phase of glucose C utilization, and not initial uptake, which was studied already many times.**

**Additional general comments**

The statistics need to be further developed. The estimates of the turnover for the different fractions/compounds (L 304) need to be described with a mean and error. R2, significance and

SE need to be added with Fig 2, 3 and 4. Current description does not make it possible to verify

the assertion of the authors that the turnover rates of the various pools are significantly different. Fig. 5 does not add to understanding or interpretation of the results and can be removed.

**Thank you for the comment. Significance and SE are presented in the figures 2, 3 and 4. Figure 5 is a synthesis of the ms results, however, if reviewer insists we will remove it.**

The observation that the 13C incorporation (as a percentage) was higher in PLFA than in cytosol does not logically result in a conclusion that the incorporation is faster (L32). This result may just be a reflection of the size of the pool (PFLA versus "cytosol"), and certainly

Удалено: ¶

does not show "the importance" of membranes "for initial C utilization".

**Thank you for the comment. On the L32 we speak about enrichment of pool by  ${}^{13}C$  - means % of  ${}^{13}C$  in the total pool C, and this completely account for the pool size. The incorporation of  ${}^{13}C$  does not account for the pool size, but we do not speak in L32 about that.**

The use of the term filamentous organisms should be avoided. The authors probably mean fungi. I like the intent of L46, however, the comparison of the dynamic behavior of the three pools remains poorly developed.

**Thank you for the comment. The term filamentous organisms can not be avoided, because we speak about fungi and actinomycetes (which can not be named as fungi). The dynamic behavior of the three pools was improved.**

Careless use of references: L 68: Malik et al have not reported on cytosol, nor on its supposed heterogeneity. It is not at all clear how location would affect the turnover time of membranes and cell walls (L70).

**Thank you for the comment. Malik et al investigated pool extracted after chloroform fumigation, which is partly reflect the composition of a cytosol pool. He measured molecular mass distribution and found out many fractions with different molecular weights, which is an observation of fraction heterogeneity. The term location was corrected to the term "function".**

Bremer and Kuikman (1993; L 73) are not experts in microbial physiology, and therefore not an authorative source to support the statement that labeled glucose appears first in the "cytosol". In fact, they only looked at the cytosol (fumigation-extractable) so cannot comment on whether other compounds or fractions become labeled first or later.

**Thank you for the comment. We provided other reference: Gottschalk, G.: Bacterial Metabolism, Springer-Verlag New York, New York, 1979.**

A reference is needed to support the assumption that "the cytosol is considered to be the most dynamic pool within microbial cells". Furthermore, heterogeneity (L75-76) has never stopped any calculation of turnover times, as is evident in soil organic matter turnover studies. Important references are missing for example those by Malik et al 2015 where comparison between "cytosol" and PFLAs are made (and DNA/RNA).

**Thank you for the comment. The reference Malik et al 2015 was added. L75-76 we mean, that due to many different compounds are in the composition of cytosol (with different molecular masses) the turnover time of C in this pool is a mean of turnover times of these components. We have corrected: Organic compounds that are taken up by microorganisms first enter the cytosol (Bremer and Kuikman, 1994), which has a high heterogeneity in composition.**

L96 and following: This paragraph tries to distinguish between cellular turnover – I assume as a consequence of cell death is what is referred to here – and turnover of compounds within a

living cell. However, it is not that easy to make that distinction – how does one distinguish between lipids being recycled and reused, taken apart and made into for example amino-acids,

Удалено: ¶

while other amino acids are recycled into lipids, and what happens after cell death – uptake of lipids by other organisms intact incorporated, reused, recycled, taken apart and/or turned into CO2. Moreover, the observations of increasing 13C concentrations for fungi versus decreasing

ones in bacteria suggests some transfer of compounds, but remains unexplained in this manuscript.

**Thank you for the comment. We corrected:**

Bacteria and fungi have various chemical composition, which strongly contributes to their turnover rates in soil: for bacteria it consists 2.3-33 days, whereas for fungi it accounts for 130-150 days (Moore et al., 2005; Rousk and Baath, 2007; Waring et al., 2013). Despite turnover of microorganisms directly effect the C turnover rates in intercellular compounds (cell membrane and cell wall biomarkers), this relationship has rarely been investigated so far. However, the comparison of C turnover for cell membrane and cell wall components can be used to characterize the contribution of various microbial groups to medium-term C utilisation and to the stabilization of microbially derived C in SOM.

L 146: it is not clear to me why unlabeled glucose was added to the control treatments.

**Thank you for the comment. Unlabelled glucose was added to make the experimental conditions equal (C content of the additions). If we add only water, than conditions of the experiment would not be the same and we would not have true controls.**

L 149: explain why the shelter was put in place and why it was removed. What was the effect of this on the soil moisture content?

**Thank you for the comment. Shelter were putted to avoid the rain and the flow of glucose out the soil columns. The soil moisture remained nearly constant. The shelter were removed because this was field experiment and natural conditions should present on it.**

L 157: why was the soil stored at 5 °C for 5 days prior to chloroform-fumigation analysis? What happened to the "cytosol" during that time? Does this mean that the value for cytosol is really the value after 8, 15, and 55 days?

**Thank you for the comment. This is standard storage temperature for the soil sample before the analysis, if they can not be performed in the same day as sampling. Due to temperature in the field was around 15°C, the storage at 5°C will not cause strong effect on the cytosol fraction.**

L 180: defumigation is not a word.

**Thank you for the comment. Was changed to: "After removing the rest of chloroform from the soil....".**

L 186: "extraction efficiency" not "extraction factor"

**Thank you for the comment. Was changed.**

L 247: "the assignment of fatty acids to microbial groups ..." is confusing me. Does this mean

Удалено: ¶

that as part of this study, biomarker PFLAs are assigned to group independent of what is done in other studies?

other studies?

**Thank you for the question. No, it means that we used previous studies to assign PLFAs measured in our experiment to the particular microbial groups. The appropriate reference is provided (Zelles, 1997).**

L 249: this procedure is not clear to me, but I am not at all familiar with PFLA/Microbial community analysis. My first impression was that the analysis is basically a community analysis – showing, based on PFLAs, what the community looks like. However, L 247 suggests that with this procedure, PFLA are assigned to microbial taxa, but then in the heading of Supplementary Table 1 it suggests that literature data is used – Please clarify what the table is used for, how (and

what) literature data is used, and what the results of this analysis means for your experiment. Similarly, L 431: the arguments for using the 16:1w5 as a biomarker for VAM and not G- are weak. The abundance of VAM needs to be expressed relative to G- bacteria. Table one suggest

that the total C for PFLAs is higher than for VAM, thus is more abundant (?).

**Thank you for the comment. The procedure is the following: PLFA analysis provides content of various PLFAs, some of them are specific for the groups of G+,G- bacteria, fungi, actinomycetes and arbuscular mycorrhizal fungi. This specifity based on the analysis of pure cultures (Zelles, 1997). Factor analysis with the principal component extraction method of mass % of individual PLFAs was done. The final assignment of fatty acids to distinct microbial groups was made by combination the results of factor loadings table with databases about presence of particular fatty acids in microbial groups (Zelles, 1997). Fatty acids which were loaded into the same factor with the same sign (+ or -) and belonged to one group (base of the table provided in Zelles (1997)) were related to one specific microbial groups and their PLFA contents were summed. This method enables quality separation of microbial groups within the soils (Apostel et al., 2013; Gunina et al., 2014).**

With PLFA analysis is not possible to conclude about the abundance of VAM and Gbacteria. Only the approximate calculated coefficient was proposed in literature (Baath and Anderson, 2003) for fungal biomass. That is why we spoke only about the content of biomarkers in our study, and not about biomass of microorganisms. Based on the PLFA-C it is not possible to conclude about the biomass of microbial groups.

L290: the description of the results (declines between 3 and 10 days but then remains constant

then constant) does not match the assumed exponential decline. Please explain.

**Thank you for the comment. The information provided at the L290 is about recovery of the tracer in various pools, whereas exponential decline is seen for the enrichment of C in the pools (portion of  ${}^{13}$ C in the total C pool).**

Fig. 2, 3, 4: the statistical tests should also be done between harvests, not only between microbial groups.

**Thank you for the comment. The statistical tests were added, where the differences between sampling points are clearly seen.**

Удалено: ¶

L 347: the explanation for the differences between this study and published results, namely the

amount of glucose added and the microbial activity, are not revealed. Some further information

on these explanatory variables would be appreciated. Is microbial activity measured in this study, microbial activity is not measured? The idea that microbes store glucose when added in

small quantities is unproven – it is a mere assumption, recently defended by Sinsabaugh et al 2013, but evidence for storage was absent in recent experiments by Dijkstra et al (2015). The idea that the storage leads to maintenance is in contradiction to the 80% recovery after 50 days,

and with the idea that microbial pools and cells turn over fast.

**Thank you for the comment. We have removed this paragraph from the ms.**

L 362: the description of the two stages of glucose decomposition -1) CO2 production plus biosynthesis, and 2) C incorporated in microbial cells is used for anabolism is confusing. Is anabolism different from biosynthesis? Is during the second phase CO2 production absent? How

do the two phases relate to the biosynthesis of lipids, cytosol, and aminosugars? Please clarify

**Thank you for the comment. We used only the term biosynthesis. We wanted to highlight that during the first phase of glucose utilization the C is mainly goes to CO2, whereas during the second phase more C goes for biosynthesis, which includes reuse of cell compounds. In our study we focused on the second phase of glucose utilization.**

L 395: what is this model, please explain some salient details and how it agrees with your observations.

**Thank you for the comment. We have removed these sentences.**

L 419: this rationalization needs some references or evidence that contact with the environment leads to rapid turnover.

**Thank you for the comment. Typical example for that is fast response of cell membranes to stress conditions (water stress, pH stress, pollution, lack of available C): formation of cyclo-fatty acids in membranes of gram-negative bacteria (Bossio et al., 1998, Guckert et al., 1986, Kieft et al., 1997).**

L 421: the problem of active and inactive cells for cytosol dynamics is similar for lipid dynamics, as inactive cells also have membranes.

**Thank you for the question. Actually the lipid behaviour is different from the cytosol pool: even inactive cells repair membranes, whereas other cell pool not necessary to be repaired. The typical example for that is: dormant microorganisms live with damaged DNA, but never with damaged membranes.**

Удалено: ¶

L 482: how is this conclusion drawn when the turnover rate cannot be calculated according to L486. L 506: how do you determine that the turnover of the amino-sugars is higher than that of

the cytosol pool? L 509: this would be a wonderful conclusion, but it does not appear in the abstract at all. What is the reason that the cytosol is so stable? Please elaborate.

**Thank you for the comment. Due to C replacement in amino sugars pool was still increasing after 50 days, we can not calculate the C turnover times. However, if replacement was still increasing, it is obviously that C turnover in amino sugars is slower than in PLFA and cytosol (where 13C replacement decreased with time). The conclusion about: " This reflects that microbial C turnover is a phenomenon that is not restricted to the death or growth of new cells, but that even within living cells, highly polymeric cell compounds, including cell walls, are constantly replaced and renewed" was included into the Abstract. The reason that cytosol is stable, is that it contains compounds with different molecular size (Malik et al., 2013): low molecular weight compounds, which renew fast and high molecular weight compounds which renew slow.**

L 511 and following – the results from the measurements seem to indicate contrasting conclusions – bacteria or fungi are most important (L516 and following). It is then stated that only the galactosamine/muramic acid ratio should be used. So, this means that the reader has wasted a number of valuable brain cells thinking about the galactosamine/glucosamine ratios, and looked at the data, but that was all a waste of time? Why not start with what is known (galac/muramic ratio) and leave it at that. Furthermore, there is a lot more text about the three aminosugars and their ratios in relation to bacteria and fungi – is that still relevant inlight of L 521?

**Thank you for the comment. We agree with reviewer, this part of discussion was reduced, and only galactosamin/muramic ratio was presented.**

Fig 1: explain what is total 13C remaining, what is non-specified pool? Remake the Fig so that

the SE of the aminosugars are fully shown.

**Thank you for the comment. Total 13C remaining is an amount of 13C measured in bulk soil, it is a sum of 13C in cytosol, PLFAs, amino sugars and non-specified pool. Non-specified pool is amount of 13C in the soil without 13C incorporated into cytosol pool, PLFAs and amino sugars. This explanation was added into the figure legend.**

Fig. 2: what is the equation with the word "replacement" in it? I think it is just the function of 13C over time, and thus the word replacement can be removed, but I may be wrong. Add R2, P value and significance (and SE of the turnover estimate)

**Thank you for the comment. This is enrichment, this is misprinting. This was corrected in the paper.**

Fig. 3: instead of showing differences between microbial groups, we need to know the differences between dates AND microbial groups to evaluate how these differences represent significant differences in turnover, and whether this turnover differs between groups. Moreover, the goal of this paper was to determine differences in turnover between microbial groups, but this is not calculated. If turnover cannot be calculated for groups where 13C

**enrichment is increasing over time, what was the basis for the conclusion that turnover differed between fungi and bacteria (L320)?**

**Thank you for the comment. The differences between dates were provided. Turnover of the different groups can be calculated only for one, which have decrease in 13C replacement. Conclusion about the differences in turnover time of C between fungi and bacteria is made based on the trend of 13C replacement: if 13C replacement was still increasing it means, that the C turnover in particular PLFAs was longer compare to one where 13C replacement was decreasing within the experimental time.**

**Fig 5: not really helpful.**

**Thank you for the comment. Fig. 5 was strongly improved: fluxes were clearly marked by the different size of arrows, position of x and y axises were changed. However, if reviewer insist we will remove it.**

Удалено: ¶

[revised manuscript text omitted]
             | I                                                                        |                                                    |     |                          |
| 185 | The cytosolic pool v              | vas determined by the fumigation-extraction                              | on technique from fresh soi                        | 1   |                          |
| 186 | shortly after samplin             | g, according to Wu et al. (1990) with slight                             | t changes. Briefly, 15 g fresl                     | 1   |                          |
| 187 | soil was placed into              | glass vials, which were exposed to chlor                                 | roform during 5 days. Afte                         | r   |                          |
| 188 | removing the rest of              | chloroform from the soil, the cytosolic C                                | was extracted from the soi                         | 1   | Удалено: defumigation    |
| 189 | with 45 mL 0.05 M                 | K 2 SO 4 . As fumigation-extraction techniq | ue allows to obtain not only                       | y   | Удалено: Due to   |
| 190 | soluble components,               | but also cell organelles and cell particle                               | es, we named pool of C in                          | 1   |                          |
| 191 | fumigated extracts                | as cytosol only for simplification of te                                 | rminology. Organic C wa                            | S   |                          |
| 192 | measured with a high              | gh-temperature combustion TOC-analyser                                   | (Analyser multi N/C 2100                           | ,   |                          |
| 193 | Analytik Jena, Gern               | nany). The cytosolic pool was calculated                                 | as the difference between                          | 1   |                          |
| 194 | organic C in fumigate             | ed and unfumigated samples without correc                                | ting for extraction efficiency                     |     | Удалено: factor.         |
| 195 | After organic C conc              | centrations were measured, the $K_2SO_4$ extra                           | cts were freeze-dried and the                      | e   |                          |
| 196 | $\delta^{13}C$ values of a 30-    | -35 µg subsample were determined using                                   | EA-IRMS (instrumentation                           | n   |                          |
| 197 | identical to soil $\delta^{13}$ C | determination). The recovery of glucose of                        | lerived 13 C in fumigated and           | d / |                          |
| 198 | unfumigated samples               | s was calculated according to the above-me                               | ntioned mixing model (Eq.                          | 1   | Удалено: uations         |
| 199 | and 2). The $^{13}$ C in          | the microbial cytosol was calculated fr                                  | om the difference in these                         | e   | VARABENO: incorporations |
| 200 | recoveries,                       |                                                                          |                                                    |     |                          |
| 201 |                                   |                                                                          |                                                    |     | Удалено: ¶               |
|     |                                   |                                                                          |                                                    |     |                          |
| 202 | 2.4. Phospholipid fat             | ty acid analysis                                                         |                                                    |     |                          |
|     |                                   |                                                                          |                                                    | /   | Удалено: ¶
¶          |

[revised manuscript text omitted]

|                                                                                                                                                           | Amino sugars were the largest pool, due to their accumulation in SOM, whereas pools that                                                                                                                                                                                                                                                                                                                                                                                                                                                                                                                                                                                                                                                                                                                          |                                                                                                                                                                                                                                              |
|-----------------------------------------------------------------------------------------------------------------------------------------------------------|-------------------------------------------------------------------------------------------------------------------------------------------------------------------------------------------------------------------------------------------------------------------------------------------------------------------------------------------------------------------------------------------------------------------------------------------------------------------------------------------------------------------------------------------------------------------------------------------------------------------------------------------------------------------------------------------------------------------------------------------------------------------------------------------------------------------|----------------------------------------------------------------------------------------------------------------------------------------------------------------------------------------------------------------------------------------------|
| 298                                                                                                                                                       | mainly characterize living MB showed smaller C contents (Table 1). The cytosolic pool (C                                                                                                                                                                                                                                                                                                                                                                                                                                                                                                                                                                                                                                                                                                                          |                                                                                                                                                                                                                                              |
| 299                                                                                                                                                       | content 210±7.10 for day 3; 195±14.8 for day 10; 198±19.9 mg C kg -1 soil for day 50) as well                                                                                                                                                                                                                                                                                                                                                                                                                                                                                                                                                                                                                                                                                                          |                                                                                                                                                                                                                                              |
| 300                                                                                                                                                       | as nearly all PLFA groups (Suppl. Table 2) remained constant during the experiment.                                                                                                                                                                                                                                                                                                                                                                                                                                                                                                                                                                                                                                                                                                                               |                                                                                                                                                                                                                                              |
| 301                                                                                                                                                       | [Table 1]                                                                                                                                                                                                                                                                                                                                                                                                                                                                                                                                                                                                                                                                                                                                                                                                         |                                                                                                                                                                                                                                              |
| 302                                                                                                                                                       |                                                                                                                                                                                                                                                                                                                                                                                                                                                                                                                                                                                                                                                                                                                                                                                                                   |                                                                                                                                                                                                                                              |
|                                                                                                                                                           |                                                                                                                                                                                                                                                                                                                                                                                                                                                                                                                                                                                                                                                                                                                                                                                                                   | Удалено: cytosolic pool                                                                                                                                                                                                                      |
| 303                                                                                                                                                       | The highest recovery of 13 C was found for cytosol pool (15–25% of applied 13 C),                                                                                                                                                                                                                                                                                                                                                                                                                                                                                                                                                                                                                                                                                                           | Удалено: amount                                                                                                                                                                                                                              |
| 304                                                                                                                                                       | whereas the lowest amount was recovered in amino sugars (0.8–1.6% of applied $^{13}$ C) (Fig. 1).                                                                                                                                                                                                                                                                                                                                                                                                                                                                                                                                                                                                                                                                                                                 | Удалено: among the investigated microbial pools                                                                                                                                                                                       |
| 305                                                                                                                                                       | The recovery of glucose derived 13 C in the cytosolic pool decreased over time, with the                                                                                                                                                                                                                                                                                                                                                                                                                                                                                                                                                                                                                                                                                                        | Удалено: amount                                                                                                                                                                                                                              |
| 306                                                                                                                                                       | largest decrease from day 3 to day 10, and then remained constant for the following month                                                                                                                                                                                                                                                                                                                                                                                                                                                                                                                                                                                                                                                                                                                         |                                                                                                                                                                                                                                              |
|                                                                                                                                                           |                                                                                                                                                                                                                                                                                                                                                                                                                                                                                                                                                                                                                                                                                                                                                                                                                   | Удалено: total                                                                                                                                                                                                                               |
| 307                                                                                                                                                       | (Fig. 1). The 13 C recovery into PLFA was generally very low and was in the same range as                                                                                                                                                                                                                                                                                                                                                                                                                                                                                                                                                                                                                                                                                                       | Удалено: incorporation                                                                                                                                                                                                                       |
| 308                                                                                                                                                       | recovery into amino sugars (Fig. 1) The $^{13}$ C recovery in PLFA showed no clear trend                                                                                                                                                                                                                                                                                                                                                                                                                                                                                                                                                                                                                                                                                                                          | Удалено: incorporation                                                                                                                                                                                                                       |
| 500                                                                                                                                                       | recovery, into uninto sugars (116. 1). The concerning in There showed no order work                                                                                                                                                                                                                                                                                                                                                                                                                                                                                                                                                                                                                                                                                                                               | Удалено: dynamics                                                                                                                                                                                                                            |
| 309                                                                                                                                                       | between the sampling points (high standard error) (Fig. 1). In contrast, 13 C recovery in amino                                                                                                                                                                                                                                                                                                                                                                                                                                                                                                                                                                                                                                                                                                        | Удалено: during                                                                                                                                                                                                                              |
|                                                                                                                                                           |                                                                                                                                                                                                                                                                                                                                                                                                                                                                                                                                                                                                                                                                                                                                                                                                                   |                                                                                                                                                                                                                                              |
| 310                                                                                                                                                       | sugars increased two fold on the $50^{\text{th}}$ day experiment (p<0.05).                                                                                                                                                                                                                                                                                                                                                                                                                                                                                                                                                                                                                                                                                                                                 |                                                                                                                                                                                                                                              |
| <li>310</li><li>311</li>                                                                                                                         | sugars increased two fold on the 50 th day experiment (p<0.05).
[Fig. 1]                                                                                                                                                                                                                                                                                                                                                                                                                                                                                                                                                                                                                                                                                                                     |                                                                                                                                                                                                                                              |
| <li>310</li><li>311</li><li>312</li>                                                                                                             | sugars increased two fold on the 50 th day experiment (p<0.05).
[Fig. 1]                                                                                                                                                                                                                                                                                                                                                                                                                                                                                                                                                                                                                                                                                                                            |                                                                                                                                                                                                                                              |
| <li>310</li><li>311</li><li>312</li><li>313</li>                                                                                                 | sugars increased two fold on the 50 th day experiment (p<0.05).
[Fig. 1]
3.2. Turnover time of C in microbial biomass pools                                                                                                                                                                                                                                                                                                                                                                                                                                                                                                                                                                                                                                                                      |                                                                                                                                                                                                                                              |
|  <li>310</li> <li>311</li> <li>312</li> <li>313</li> <li>314</li>                                                                                | sugars increased two fold on the 50 th day experiment (p<0.05). [Fig. 1] 3.2. Turnover time of C in microbial biomass pools To evaluate C turnover in the cytosol, PLFAs and amino sugars, we calculated the                                                                                                                                                                                                                                                                                                                                                                                                                                                                                                                                                                                           |                                                                                                                                                                                                                                              |
|  <li>310</li> <li>311</li> <li>312</li> <li>313</li> <li>314</li> <li>315</li>                                                                   |  <li>sugars increased two fold on the 50th day experiment (p<0.05).</li> <li>[Fig. 1]</li> <li>3.2. Turnover time of C in microbial biomass pools</li> <li>To evaluate C turnover in the cytosol, PLFAs and amino sugars, we calculated the enrichment (% of incorporated 13C relatively to pool C) of each pool by glucose derived 13C.</li>                                                                                                                                                                                                                                                                                                                                                                                                                        | Удалено: in                                                                                                                                                                                                                                  |
|  <li>310</li> <li>311</li> <li>312</li> <li>313</li> <li>314</li> <li>315</li> <li>316</li>                                                      |  <li>sugars increased two fold on the 50th day experiment (p<0.05).</li> <li>[Fig. 1]</li> <li>3.2. Turnover time of C in microbial biomass pools</li> <li>To evaluate C turnover in the cytosol, PLFAs and amino sugars, we calculated the enrichment (% of incorporated 13C relatively to pool C) of each pool by glucose derived 13C.</li> <li>The pool enrichment was the highest for PLFAs and the lowest for amino sugars (Fig. 2)</li>                                                                                                                                                                                                                                                                                                                        | Удалено: in
Удалено: (Fig. 2)                                                                                                                                                                                               |
|  <li>310</li> <li>311</li> <li>312</li> <li>313</li> <li>314</li> <li>315</li> <li>316</li>                                                      | sugars increased two fold on the 50 th day experiment (p<0.05).                                                                                                                                                                                                                                                                                                                                                                                                                                                                                                                                                                                                                                                                                                                                        | Удалено: in
Удалено: (Fig. 2)
Удалено: was 5–8 times lower                                                                                                                                                                             |
|  <li>310</li> <li>311</li> <li>312</li> <li>313</li> <li>314</li> <li>315</li> <li>316</li> <li>317</li>                                         | sugars increased two fold on the 50 th day experiment (p<0.05).[Fig. 1]3.2. Turnover time of C in microbial biomass poolsTo evaluate C turnover in the cytosol, PLFAs and amino sugars, we calculated the
enrichment (% of incorporated 13 C relatively to pool C) of each pool by glucose derived 13 C.The pool enrichment _was the highest for PLFAs and the lowest for amino sugars (Fig. 2).Based on the decrease of 13 C enrichment over time (Fig. 2), the C turnover in the cytosol and                                                                                                                                                                                                                                                                     | Удалено: in
Удалено: (Fig. 2)
Удалено: was 5–8 times lower
in
Удалено: cytosolic pool. The
13 C enrichment in                                                                                                      |
|  <li>310</li> <li>311</li> <li>312</li> <li>313</li> <li>314</li> <li>315</li> <li>316</li> <li>317</li> <li>318</li>                            |  <li>sugars increased two fold on the 50th day experiment (p<0.05).</li> <li>[Fig. 1]</li> <li>3.2. Turnover time of C in microbial biomass pools</li> <li>To evaluate C turnover in the cytosol, PLFAs and amino sugars, we calculated the enrichment (% of incorporated 13C relatively to pool C) of each pool by glucose derived 13C.</li> <li>The pool enrichment was the highest for PLFAs and the lowest for amino sugars (Fig. 2).</li> <li>Based on the decrease of 13C enrichment over time (Fig. 2), the C turnover in the cytosol and PLFAs was calculated as 151 and 47 days, respectively. The C turnover time in the amino-</li>                                                                                                            | Удалено: in         Удалено: (Fig. 2)         Удалено: was 5–8 times lower in         Удалено: cytosolic pool. The 13 C enrichment in         Удалено: was the lowest                                                             |
|  <li>310</li> <li>311</li> <li>312</li> <li>313</li> <li>314</li> <li>315</li> <li>316</li> <li>317</li> <li>318</li>                            | sugars increased two fold on the 50 th day experiment (p<0.05). [Fig. 1] 3.2. Turnover time of C in microbial biomass pools To evaluate C turnover in the cytosol, PLFAs and amino sugars, we calculated the enrichment (% of incorporated 13 C relatively to pool C) of each pool by glucose derived 13 C. The pool enrichment _was the highest for PLFAs and the lowest for amino sugars (Fig. 2). Based on the decrease of 13 C enrichment over time (Fig. 2), the C turnover in the cytosol and PLFAs was calculated as 151 and 47 days, respectively. The C turnover time in the amino-                                                                                                                                                                          | Удалено: in
Удалено: (Fig. 2)
Удалено: was 5–8 times lower
in
Удалено: cytosolic pool. The
13 C enrichment in
Удалено: was the lowest
Удалено: incorporation                                                 |
|  <li>310</li> <li>311</li> <li>312</li> <li>313</li> <li>314</li> <li>315</li> <li>316</li> <li>317</li> <li>318</li> <li>319</li>               | sugars increased two fold on the 50 th day experiment (p<0.05). [Fig. 1] 3.2. Turnover time of C in microbial biomass pools To evaluate C turnover in the cytosol, PLFAs and amino sugars, we calculated the enrichment (% of incorporated 13 C relatively to pool C) of each pool by glucose derived 13 C. The pool enrichment was the highest for PLFAs and the lowest for amino sugars (Fig. 2). Based on the decrease of 13 C enrichment over time (Fig. 2), the C turnover in the cytosol and PLFAs was calculated as 151 and 47 days, respectively. The C turnover time in the aminosugar pool could not be calculated by this approach because the maximum enrichment had                                                                                      | Удалено: in         Удалено: (Fig. 2)         Удалено: was 5–8 times lower in         Удалено: cytosolic pool. The I 3 C enrichment in         Удалено: was the lowest         Удалено: incorporation                             |
|  <li>310</li> <li>311</li> <li>312</li> <li>313</li> <li>314</li> <li>315</li> <li>316</li> <li>317</li> <li>318</li> <li>319</li> <li>320</li>  | sugars increased two fold on the 50 th day experiment (p<0.05). [Fig. 1] 3.2. Turnover time of C in microbial biomass pools To evaluate C turnover in the cytosol, PLFAs and amino sugars, we calculated the enrichment (% of incorporated 13 C relatively to pool C) of each pool by glucose derived 13 C. The pool enrichment was the highest for PLFAs and the lowest for amino sugars (Fig. 2). Based on the decrease of 13 C enrichment over time (Fig. 2), the C turnover in the cytosol and PLFAs was calculated as 151 and 47 days, respectively. The C turnover time in the aminosugar pool could not be calculated by this approach because the maximum enrichment had not yet been reached, and consequently a decomposition function could not be fitted. | Удалено: in         Удалено: (Fig. 2)         Удалено: was 5–8 times lower in         Удалено: cytosolic pool. The 13 C enrichment in         Удалено: was the lowest         Удалено: incorporation         Удалено: ¶         ¶ |

321

**[Fig. 2]**

| 322                                                                                                                                          |                                                                                                                                                                                                                                                                                                                                                                                                                                                                                                                                                                                                                                                                                                    |                                                                                                                                      |
|----------------------------------------------------------------------------------------------------------------------------------------------|----------------------------------------------------------------------------------------------------------------------------------------------------------------------------------------------------------------------------------------------------------------------------------------------------------------------------------------------------------------------------------------------------------------------------------------------------------------------------------------------------------------------------------------------------------------------------------------------------------------------------------------------------------------------------------------------------|--------------------------------------------------------------------------------------------------------------------------------------|
| 323                                                                                                                                          | 3.3 Phospholipid fatty acids                                                                                                                                                                                                                                                                                                                                                                                                                                                                                                                                                                                                                                                                       |                                                                                                                                      |
| 324                                                                                                                                          | Fatty acids of bacterial origin dominated over those of fungal origin within the living                                                                                                                                                                                                                                                                                                                                                                                                                                                                                                                                                                                                            |                                                                                                                                      |
| 325                                                                                                                                          | microbial community characterized by PLFA composition (Table 1), The PLFA content of                                                                                                                                                                                                                                                                                                                                                                                                                                                                                                                                                                                                               | Удалено: Gram-negative (G-)
fatty acids were more abundant
than gram-positive (G+) ones.
Actinomycetes and vesicular |
| 326                                                                                                                                          | most groups did not change significantly during the experiment, reflecting steady-state                                                                                                                                                                                                                                                                                                                                                                                                                                                                                                                                                                                                            | arbuscular mycorrhiza (VAM)
fatty acids dominated in the                                                                          |
| 327                                                                                                                                          | conditions for the microbial community (see Suppl , Table 2).                                                                                                                                                                                                                                                                                                                                                                                                                                                                                                                                                                                                                               | composition of filamentous
microorganisms, and saprotrophic
fungi showed a relatively low
presence in PLFAs                 |
| 328                                                                                                                                          | Glucose derived 13 C was incorporated in higher portions into bacterial than into fungal                                                                                                                                                                                                                                                                                                                                                                                                                                                                                                                                                                                                | Удалено: supplementary                                                                                                               |
| 329                                                                                                                                          | PLFAs (Fig. 3, top). Remarkably, the 13 C enrichment decreased over time for all bacterial                                                                                                                                                                                                                                                                                                                                                                                                                                                                                                                                                                                              |                                                                                                                                      |
| 330                                                                                                                                          | PLFAs, whereas it increased or remained constant for 16:105, fungi and filamentous,                                                                                                                                                                                                                                                                                                                                                                                                                                                                                                                                                                                                                | Удалено: VAM                                                                                                                         |
| 331                                                                                                                                          | bacterial actinomycetes (Fig. 3, bottom), indicating differences in C turnover in single-celled                                                                                                                                                                                                                                                                                                                                                                                                                                                                                                                                                                                                    |                                                                                                                                      |
| 332                                                                                                                                          | organisms compared to filamentous organisms.                                                                                                                                                                                                                                                                                                                                                                                                                                                                                                                                                                                                                                                       |                                                                                                                                      |
| 333                                                                                                                                          | [Fig. 3]                                                                                                                                                                                                                                                                                                                                                                                                                                                                                                                                                                                                                                                                                           |                                                                                                                                      |
|                                                                                                                                              |                                                                                                                                                                                                                                                                                                                                                                                                                                                                                                                                                                                                                                                                                                    |                                                                                                                                      |
| 334                                                                                                                                          |                                                                                                                                                                                                                                                                                                                                                                                                                                                                                                                                                                                                                                                                                                    |                                                                                                                                      |
| 334
335                                                                                                                                   | 3.4. Amino sugars                                                                                                                                                                                                                                                                                                                                                                                                                                                                                                                                                                                                                                                                                  |                                                                                                                                      |
| <li>334</li><li>335</li><li>336</li>                                                                                                | 3.4. Amino sugars
The content of amino sugars followed the order: muramic acid < galactosamine <                                                                                                                                                                                                                                                                                                                                                                                                                                                                                                                                                                                                |                                                                                                                                      |
| <li>334</li><li>335</li><li>336</li><li>337</li>                                                                                    | 3.4. Amino sugars
The content of amino sugars followed the order: muramic acid < galactosamine < glucosamine (Table 1). The glucosamine/muramic acid ratio varied between 17 and 55,                                                                                                                                                                                                                                                                                                                                                                                                                                                                                                     |                                                                                                                                      |
|  <li>334</li> <li>335</li> <li>336</li> <li>337</li> <li>338</li>                                                                   | 3.4. Amino sugars
The content of amino sugars followed the order: muramic acid < galactosamine < glucosamine (Table 1). The glucosamine/muramic acid ratio varied between 17 and 55, whereas the galactosamine/muramic acid ratio ranged between 12 and 19 (Table 1). This                                                                                                                                                                                                                                                                                                                                                                                                               |                                                                                                                                      |
|  <li>334</li> <li>335</li> <li>336</li> <li>337</li> <li>338</li> <li>339</li>                                                      | 3.4. Amino sugars
The content of amino sugars followed the order: muramic acid < galactosamine < glucosamine (Table 1). The glucosamine/muramic acid ratio varied between 17 and 55, whereas the galactosamine/muramic acid ratio ranged between 12 and 19 (Table 1). This provides evidence that bacterial residues were dominant in the composition of microbial                                                                                                                                                                                                                                                                                                                       |                                                                                                                                      |
|  <li>334</li> <li>335</li> <li>336</li> <li>337</li> <li>338</li> <li>339</li> <li>340</li>                                         | 3.4. Amino sugars
The content of amino sugars followed the order: muramic acid < galactosamine < glucosamine (Table 1). The glucosamine/muramic acid ratio varied between 17 and 55, whereas the galactosamine/muramic acid ratio ranged between 12 and 19 (Table 1). This provides evidence that bacterial residues were dominant in the composition of microbial residues in SOM.                                                                                                                                                                                                                                                                                                      |                                                                                                                                      |
|  <li>334</li> <li>335</li> <li>336</li> <li>337</li> <li>338</li> <li>339</li> <li>340</li> <li>341</li>                            | 3.4. Amino sugars
The content of amino sugars followed the order: muramic acid < galactosamine < glucosamine (Table 1). The glucosamine/muramic acid ratio varied between 17 and 55, whereas the galactosamine/muramic acid ratio ranged between 12 and 19 (Table 1). This provides evidence that bacterial residues were dominant in the composition of microbial residues in SOM.
The recovery, of glucose derived 13 C into amino sugars increased in the order:                                                                                                                                                                                                        | Удалено: incorporation                                                                                                               |
|  <li>334</li> <li>335</li> <li>336</li> <li>337</li> <li>338</li> <li>339</li> <li>340</li> <li>341</li> <li>342</li>               | 3.4. Amino sugars
The content of amino sugars followed the order: muramic acid < galactosamine < glucosamine (Table 1). The glucosamine/muramic acid ratio varied between 17 and 55, whereas the galactosamine/muramic acid ratio ranged between 12 and 19 (Table 1). This provides evidence that bacterial residues were dominant in the composition of microbial residues in SOM.
The recovery, of glucose derived 13 C into amino sugars increased in the order: muramic acid = galactosamine < glucosamine (Fig. 4, top) reflecting partly their pool sizes.                                                                                                           | Удалено: incorporation                                                                                                               |
|  <li>334</li> <li>335</li> <li>336</li> <li>337</li> <li>338</li> <li>339</li> <li>340</li> <li>341</li> <li>342</li> <li>343</li>  | 3.4. Amino sugars
The content of amino sugars followed the order: muramic acid < galactosamine < glucosamine (Table 1). The glucosamine/muramic acid ratio varied between 17 and 55, whereas the galactosamine/muramic acid ratio ranged between 12 and 19 (Table 1). This provides evidence that bacterial residues were dominant in the composition of microbial residues in SOM.
The recovery, of glucose derived 13 C into amino sugars increased in the order: muramic acid = galactosamine < glucosamine (Fig. 4, top) reflecting partly their pool sizes.
The 13 C recovery, showed no increase from day 3 to day 50 for any amino sugars. The ratios | Удалено: incorporation                                                                                                               |
|  <li>334</li> <li>335</li> <li>336</li> <li>337</li> <li>338</li> <li>339</li> <li>340</li> <li>341</li> <li>342</li> <li>343</li>  | 3.4. Amino sugars
The content of amino sugars followed the order: muramic acid < galactosamine < glucosamine (Table 1). The glucosamine/muramic acid ratio varied between 17 and 55, whereas the galactosamine/muramic acid ratio ranged between 12 and 19 (Table 1). This provides evidence that bacterial residues were dominant in the composition of microbial residues in SOM.
The recovery, of glucose derived 13 C into amino sugars increased in the order: muramic acid = galactosamine < glucosamine (Fig. 4, top) reflecting partly their pool sizes.
The 13 C recovery, showed no increase from day 3 to day 50 for any amino sugars. The ratios | Удалено: incorporation         Удалено: incorporation         Удалено: ¶         ¶                                                   |
|  <li>334</li> <li>335</li> <li>336</li> <li>337</li> <li>338</li> <li>339</li> <li>340</li> <li>341</li> <li>342</li> <li>343</li>  | 3.4. 
[revised manuscript text omitted]

in contrast to cytosol and
pools. |
| 170 | are porymers that require a rather complex closyfitheois of the animo sugar freeds, 2) cen wan                                                                                                 |           | .                                                              |
| 477 | polymerization occurs extracellularly (Lengeler et al., 1999) and 3) microorganisms do not                                                                                                     |           |                                                                       |
| 478 | need to synthesize peptidoglycan unless they multiply. To calculate C turnover time in this                                                                                                    | 1         | Удалено: further                                                      |
| 479 | pool, conducting of long-term experiments is necessary.                                                                                                                                        |           | Удалено: sampling poi
13 C-amino-sugar analysis  |
| 480 | The majority of amino sugars extracted after acid hydrolysis represent microbial                                                                                                               |           |                                                                       |
| 481 | necromass, which does not incorporate any glucose derived 13 C, but strongly dilutes the 13 C                                                                            | ,         |                                                                       |
| 482 | incorporated into the walls of living cells. To estimate the 13 C recovery into amino sugars of                                                                                     |           | Удалено: incorporation                                                |
| 483 | living cells, we first calculated the amount of amino sugars in the living MB pool based on                                                                                                    |           |                                                                       |

484 the fatty acids content. Assuming that PLFAs are present only in living biomass, and that the ratio of fatty acids to amino sugars in living biomass is about 0.23 (Lengeler et al., 1999), we 485

estimated the amount of amino sugars in living MB to be 0.20  $\mu$ mol g-1 soil fatty acids/0.23 = 486

0.87 µmol g-1 soil. The estimated percentage of amino sugars in living biomass from the total 487

amino sugar pool was 0.87/7.70 (total AS (µmol g-1 soil))\*100 = 11%. This estimate agrees 488 36

(од поля изменен Этформатировано: нглийский (США) /далено: incorporation of

/далено: . The total 13С corporation **Далено:** enrichment of amino ugars

Далено: from day 3 to day 50, contrast to cytosol and PLFA ools.

[revised manuscript text omitted]

---

## Editor Decision (ED1)

[revised manuscript text omitted]

---

## Author Response (AR2)

**Определение стиля:** Знак
Знак: Шрифт: курсив

Revision of the Ms. No. Biogeosciences Discuss., doi:10.5194/bg-2016-214, 2016
**Glucose C turnover in cell compartments and microbial groups in soil**

Dear Prof. Pendall, we are very thankful for your helpful suggestions and recommendations. We carefully improved the ms according to your comments and incorporated nearly all your suggestions. Please find below the responses to the comments (in green) and the improved ms.
We hope that the ms fulfill the requirements of Biogeosciences.

With kind regards,
Anna Gunina and co-authors.

**Editor review**
**The manuscript has been improved by streamlining the hypotheses and reducing the speculations and assumptions presented in the original version. However, some revisions are still required before the paper can be published. Grammar is still problematic in some sections. See attached pdf for examples of corrections.**

- Thank you for the comment. We have corrected the grammar in the ms, and paid the most attention to Abstract and conclusions sections.

**Abstract: The abstract should be revised to improve the English and also to make the most important points come across more strongly.**

- Thank you for the comment. Abstract was improved, please, see the corrected  version provided below.

**The authors need to reconcile the terminology to be consistent with the methods, particularly related to the concept of "turnover time".**

- Thank you for the comment. We have corrected the terminology regarding the "turnover time", and made necessary corrections, namely: we corrected equations, and statements in the discussion section.

**Line 35: What is meant by "renewal"?**

- Thank you for the comment. We wanted to stress that C in a pool of PLFA or amino sugars is replaced by the new $^{13}$C (from added glucose), means that C is renewed. The abstract was changed and this term was excluded.

**Lines 275-277: what is meant by "per column"? Maybe you mean "per component"?**

- Thank you for the comment.  All calculations were done to the weight of all soil in the experimental unit, which was "column" in our case. Due to we have collected the columns completely (means all soil where $^{13}$C was added was collected during sampling), we calculated the amount of $^{13}$C to the weight of all soil in the column, which was 1.5 kg (this information presented in the materials and methods section).

¶

**Lines 304-306 and throughout: Cytosol pool vs cytosolic pool: Be consistent, choose one.**

- Thank you for the comment.  We have corrected and used only "cytosol" abbreviation.

**Lines 308-309 and throughout: "recovery in" or "incorporation into"**

- Thank you for the comment.  We have checked through the all ms. Term "recovery" is correct. Only in one case L417 the term incorporation is used, due to there we spoke about incorporation of C into PLFA pool (not from pulse-labelling, but due to the uptake of C which is naturally in the soil).

**Lines 373-377: How was the "non-specific pool" of SOM determined?**

- Thank you for the comment.  The $^{13}$C in the non-specified SOM was calculated by subtracting off total $^{13}$C measured in the soil, the $^{13}$C incorporated into cytosol, PLFAs and amino sugars. This information was added into materials and method section.

**Lines 481-502: This paragraph is long and rambling and presents some results and calculations in addition to a discussion point. It would benefit the manuscript if the calculations could be put into a supplement and then the main point of the paragraph would be more clear (that amino sugars are slower to incorporate C than the other cell components studied).**

- Thank you for the comment.  We have put the calculations into the supplementary materials.

**Conclusions; This section reads like a lengthy summary of the results. It could be shortened to the main implications of the study.**

- Thank you for the comment. The conclusion section was shortened, please see the corrected version of ms, presented below:
Tracing the $^{13}$C labelled glucose through cytosol, PLFAs and amino sugars is a prerequisite for understanding the fate of organic substrates in soil and can be used to estimate C turnover times in various microbial cell compartments. In contradiction to hypothesis one, the C turnover times were as follows: PLFA (47 days)<cytosol (150 days)<amino sugars. The long C half-life time in cytosol can be explained by efficient C recycling and cytosol heterogeneous composition, which involves compounds with different turnover rates. Due to significant part of amino sugar pool was in the composition of microbial residues, the $^{13}$C enrichment of this pool was still increasing at the end of the experiment, which reflects the slowest C turnover time here. An approximate calculation of $^{13}$C enrichment of amino sugars in the living biomass accounted for 0.57% of pool size, which was lower than for PLFAs. This reflects that C turnover in cell wall components is slower than in membrane components.

Both PLFAs and amino sugars analysis showed the prevalence of bacterial biomass/bacterial residues in investigated soil. Much higher recovery and enrichment by glucose-$^{13}$C was found in bacterial than in fungal PLFAs. A lower $^{13}$C enrichment of filamentous PLFAs compare to bacterial demonstrates that i) C turnover in filamentous PLFAs is slower compare to bacterial and ii) filamentous organisms might consume bacterial biomass and utilize products of its metabolism. The ratio of galactosamine/muramic acid for incorporated $^{13}$C evidences that bacteria were more active in glucose utilisation than fungi.

¶

The $^{13}$C enrichment was the highest for muramic acid and the lowest for galactosamine, demonstrating that the turnover of bacterial cell wall components is more rapid than fungal.

Consequently, the combination of $^{13}$C labeling with the subsequent analysis of several microbial cell compartments and biomarkers is a unique approach to understanding C partitioning within microbial cells and the microbial communities in soil. This knowledge is not only crucial for assessing C fluxes and recycling in soil, but is also important for estimation the contribution of C from microbial residues to SOM.

**Figures**
**Fig. 1: Please indicate what the top brown line indicates; apparently it is the 13C-enriched glucose remaining in the soil?**

- Thank you for the comment. Brown line indicates the remaining glucose-derived $^{13}$C in the soil.

**Fig. 5: Missing?**

- We have deleted this figure according to the reviewers suggestions.
We have added figure 5 for the revision by the Editor.

¶

[revised manuscript text omitted]

¶

Figure 02.

[Figure]

¶

Figure 03.

[Figure]

¶

Figure 04.

[Figure]

¶

[Figure]

¶